# NEUTAG: Graph Transformer for Attributed Graphs

**Shubham Gupta**                                                      *shubham.gupta@cse.iitd.ac.in*
*Department of Computer Science and Engineering*
*Indian Institute of Technology Delhi, New Delhi*

**Sayan Ranu**                                                         *sayanranu@cse.iitd.ac.in*
*Department of Computer Science and Engineering*
*Indian Institute of Technology Delhi, New Delhi*

**Srikanta Bedathur**                                                  *srikanta@cse.iitd.ac.in*
*Department of Computer Science and Engineering*
*Indian Institute of Technology Delhi, New Delhi*

**Reviewed on OpenReview:** *https://openreview.net/forum?id=kQrIrYvbbw*

## Abstract

Graph Transformers (GT) have demonstrated their superiority in graph classification tasks, but their performance in node classification settings remains below par. They are designed for either homophilic or heterophilic graphs and show poor scalability to million-sized graphs. In this paper, we address these limitations for node classification tasks by designing a model that utilizes a special feature encoding that transforms the input graph separating nodes and features, which enables the flow of information not only from the local neighborhood of a node but also from distant nodes, via their connections through shared feature nodes. We theoretically demonstrate that this design allows each node to exchange information with all nodes in the graph, effectively mimicking all-node-pair message passing without requiring dense attention between all node pairs. This enables scalability for large attributed graphs when the number of features is substantially smaller than the number of nodes. We further analyze the universal approximation ability of the proposed transformer. Finally, we demonstrate the effectiveness of the proposed method on diverse sets of large-scale graphs, including the homophilic & the heterophilic varieties.

## 1 Introduction

Graph neural networks (GNN) (Hamilton et al., 2017; Veličković et al., 2018; Xu et al., 2019; Abu-El-Haija et al., 2019; Nishad et al., 2021) are increasingly considered de facto models for solving graph mining tasks such as graph classification, node classification, link prediction, etc. Their utility has been demonstrated across diverse domains, including drug and material discovery Goyal et al. (2020); Bihani et al. (2023), traffic forecasting Gupta et al. (2023); Jain et al. (2021), recommendation systems Gupta et al. (2025); Sirohi et al. (2024), and modeling physical interacting systems Bishnoi et al. (2024; 2023); Bhattoo et al. (2022). Recent advances in the transformer (Vaswani et al., 2017) family of neural networks, especially in the domain of language (Devlin et al., 2019; Radford et al., 2019), and vision (Dosovitskiy et al., 2020) have propelled their applications in the graph domain as well, specifically for graph classification tasks (Rampášek et al., 2022). Graph Transformers (GT) take node sequences as input with their attributes, structural and position encoding, and apply transformer layers (Vaswani et al., 2017) successively to learn contextual node representation. It enables modeling long-range dependencies among nodes, avoids over-smoothing (Liu et al., 2020) problems linked with deeper GNN, and is more expressive due to structural (Dwivedi et al., 2022a) and position encodings (Kreuzer et al., 2021a; Dwivedi et al., 2023a). These advantages in the GT architecture

lead them to outperform other GNN-based methods, especially in molecular and biological graph classification tasks, as shown in GraphGPS (Rampášek et al., 2022).

However, the sizes of graphs considered in graph classification tasks are typically of small scale, that is, $\approx$ 100 nodes per graph. In contrast, node classification tasks often involve graphs with millions of nodes, such as snap-patents (Lim et al., 2021). One of the fundamental limitations of using graph transformers in these settings is dense attention, which computes attention over all node pairs, leading to $\mathcal{O}(N^2)$ computation per layer. This is computationally prohibitive and not practical for applications involving large graphs. Recently, sparse-attention methods (Choromanski et al., 2020; Zaheer et al., 2021) proposed in language models have been utilized in GT (Rampášek et al., 2022) to approximate dense attention. However, these sparse-attention methods don't explicitly leverage the structural properties of graphs, resulting in sub-optimal performance than dense attention-based graph transformers. Recently, graph-specific sparse transformers (Rampášek et al., 2022; Shirzad et al., 2023; Kong et al., 2023; Chen et al., 2022b) have been proposed to incorporate graph topology, which learn virtual tokens via global nodes, anchor nodes, or clustering. However, none of these methods explicitly uses the feature dimension as a carrier for long-range communication.

## 1.1 Existing Works and their limitations

A plethora of graph transformers have been proposed to target many aspects of representation learning. A recent survey (Müller et al., 2024) characterizes these key innovations into four primary dimensions: 1) the design of positional and structural encodings (Chen et al., 2022a; Dwivedi et al., 2022a; Bouritsas et al., 2021; Kreuzer et al., 2021b; Lim et al., 2023; Dwivedi & Bresson, 2021; Wang et al., 2022), 2) handling of geometric vs non-geometric features (Fuchs et al., 2020; 2021; Shi et al., 2023; Luo et al., 2023), 3) graph tokenization (Kim et al., 2022; Hussain et al., 2022; Chen et al., 2022b), and 4) propagation mechanisms (Rampášek et al., 2022; Shirzad et al., 2023; Kong et al., 2023; Chen et al., 2022b; Dwivedi et al., 2023b; Ma et al., 2023; Liu et al., 2023; Kuang et al., 2021; Zhu et al., 2024; Liu et al., 2023). We have additionally identified a fifth dimension focusing on replacing standard self-attention formulae with equivalent, scalable formulations using linear, polynomial and diffusion processes based kernels (Wu et al., 2023a;b; Deng et al., 2024). These improve scalability but at the cost of expressivity (Deng et al., 2024) and still require graph partitioning on million-sized graphs, breaking all-pair connectivity.

This paper focuses primarily on the fourth category, which aims to approximate all pair attention connectivity. These methods typically adopt a common modular architecture. Each layer is composed of a message-passing neural network (MPNN) (Hamilton et al., 2017; Veličković et al., 2018; Xu et al., 2019) followed by a transformer layer. The principal distinctions between methods lie in the design of the transformer layer. GraphGPS utilizes all-pairs attention. Exphormer combines GNN with a transformer layer consisting of local neighbor attention and non-local attention via virtual nodes. These virtual nodes are connected to every node in the graph, leading to an approximation of all-node-pairs attention. Goat (Kong et al., 2023) and LargeGT (Dwivedi et al., 2023b) removes the GNN component entirely by replacing virtual nodes in Exphormer using trainable and clustering-based virtual nodes maintained using a code-book computed by $k$-means clustering and updated through exponentially moving averages.

- **Redundant dependency on Gnn:** Hybrid GT such as GraphGPS and Exphormer utilize a modular architecture of GNN and transformers. While effective, this creates redundant dependence, despite the literature showing that Transformers with positional/structural encodings are theoretically universal approximators of graph-to-graph functions (Yun et al., 2020; Kreuzer et al., 2021a).
- **Limited to either homophilic or heterophilic graphs:** Hybrid architectures inherit the biases of the underlying GNN. If GNN derives the node representation mainly from its local connectivity, then it will propagate homophily biases in the transformer. Similarly, if a GNN is designed for heterophilic graphs that use higher-hop nodes, it will propagate non-homophily biases in the transformer. For example, Exphormer and GraphGPS use Gcn (Kipf & Welling, 2017) as GNN, which leads to their good performance on homophilic graphs (Sen et al., 2008; Yang et al., 2016) but not on heterophilic graphs (Pei et al., 2020; Rozemberczki et al., 2021; Lim et al., 2021). We show this effect empirically in Table 1, where we remove the Gcn component to show how their performance improves on heterophilic graphs but reduces on homophilic graphs.

- **Non-scalable or assumptions based:** Dense-attention and global token-based transformers (Rampášek et al., 2022; Shirzad et al., 2023) require the full graph in GPU memory, making training on large graphs challenging. Linear-attention variants (Wu et al., 2023a; Deng et al., 2024; Wu et al., 2023b) improve memory efficiency and increase scalability, but still require graph partitioning for million-node datasets, which breaks all-pairs attention and often sacrifices expressivity due to kernel approximations (Deng et al., 2024). Clustering-based virtual node methods (Kong et al., 2023; Dwivedi et al., 2023b; Zhu et al., 2024) avoid GNN reliance but require a trainable projection matrix and tuning of the number of virtual nodes.

### 1.2 Contributions

To address the gaps outlined above, we propose Neural Transformer for Attributed Graphs (NEUTAG). We use graph transformations with node features as virtual nodes to design novel sparse graph transformers. We examine the benefits of the aforementioned transformation, including increased graph homophily. In summary, NEUTAG offers the following significant advantages over the existing work.

- **Scalable and data-agnostic modeling for attributed graphs:** NEUTAG transforms the input graph into a bipartite structure consisting of graph nodes and virtual feature nodes, and sparsifies the all-pair attention into a unified attention mechanism over local neighbors and virtual feature neighbors. This allows NEUTAG to flexibly learn from both homophilic patterns via local structures and heterophilic patterns via non-local feature nodes, leading to a data-agnostic architecture. And, since its virtual nodes are based on features that are derived deterministically from the input attribute space rather than learned through a clustering mechanism, NEUTAG avoids the need for clustering-based virtual-node construction. The resulting sparse-attention mechanism seamlessly facilitates node batching and enables NEUTAG to scale to large attributed graphs when the average number of connected features per node is substantially smaller than the graph size.
- **Theoretical analysis:** We establish the theoretical grounding of the proposed transformation by proving that it increases the connectivity in the graph. Moreover, we investigate the theoretical and parameter conditions under which the NEUTAG potentially serves as a permutation-equivariant universal approximator of a dense attention layer.
- **Empirical evaluation:** We perform extensive experiments on real-world datasets, including both homophilic and heterophilic datasets, along with a large-scale dataset *snap-patent* containing 2.9 million nodes and 13.9 million edges. We evaluate our proposed method against 12 graph transformer baselines, including 15 variants. We clearly establish that the proposed sparse graph transformer NEUTAG is competitive in both homophilic and heterophilic graphs, as well as in small-scale and large-scale graphs, consistently.

**Paper organization:** Section A in appendix introduces preliminaries on graphs, graph neural networks, transformers, and graph transformers, along with notations and problem formulation. Section 3 presents our proposed methodology: we first describe the graph transformation and its benefits for structural connectivity, then introduce the attention mechanism on the transformed graph and analyze its all-pair attention approximation capabilities. Section 4 details the experimental setup, including datasets, baselines, evaluation metrics, and benchmarks NEUTAG. Finally, Section 6 concludes the paper.

## 2 Preliminaries

We provide preliminaries on Graphs, Graph Neural Networks, Transformers, and Graph Transformers, including definitions, notations, and problem formulation in Appendix A.

## 3 Methodology

This section presents two main components. First, we propose a graph transformation that decouples features from nodes into separate nodes, improving structural connectivity and increasing effective homophily, which we formally analyze. Building on this transformation, we introduce a novel attention mechanism that facilitates information flow between graph and feature nodes.

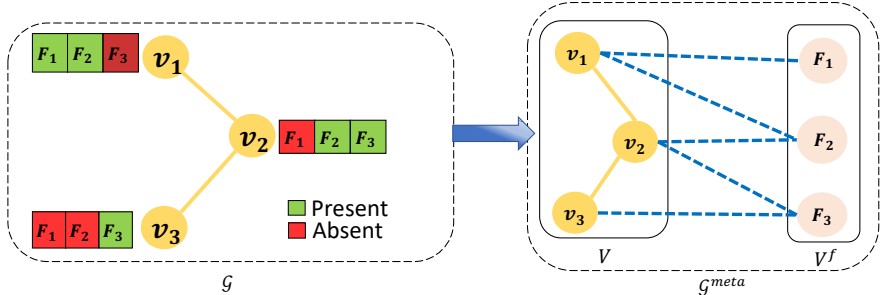

Figure 1: Transformation of input graph $\mathcal{G}$ into its metamorphosis form $\mathcal{G}^{meta}$. In $\mathcal{G}$, the features marked in green are present for the corresponding node. $\mathcal{G}^{meta}$ contains additional edges with all present features.

### 3.1 Graph transformation

Given a graph $\mathcal{G} = (\mathcal{V}, \mathcal{E}, \mathbf{X})$ and its feature set $\mathcal{F}$ as defined in def. 1 in appendix A where $\mathcal{F} = \bigcup_{v \in \mathcal{V}} \mathcal{F}_v$ is the set of all features in graph $\mathcal{G}$. $\mathcal{F}_v$ is a feature set at node $v \in \mathcal{V}$ we convert it to its metamorphosis form $\mathcal{G}^{meta} = (\mathcal{V}^{meta}, \mathcal{E}^{meta}, \mathbf{X})$ as follows. First, we create virtual feature nodes $V^f$ corresponding to feature set $\mathcal{F}$ of graph $\mathcal{G}$ and add these new nodes to $\mathcal{V}$ to create $\mathcal{V}^{meta}$. Formally,

$$\mathcal{V}^{meta} = \mathcal{V} \cup \mathcal{V}^f : \mathcal{V}^f = \{f \in \mathcal{F}\} \tag{1}$$

Next, we retain the original edges $\mathcal{E}$ in $\mathcal{G}^{meta}$ and create the edges that connect original graph nodes $\mathcal{V}^G$ and $\mathcal{V}^f$ as follows.

$$\mathcal{E}^f = \{(v, f) \mid v \in \mathcal{V}, f \in \mathcal{F}, \mathbf{X}[v, f] = 1\} \tag{2}$$

We overload the matrix index operation with the access operator $[]$. $\mathcal{E}^f$ represents the set of edges between graph nodes $\mathcal{V}$ and features which are present in corresponding nodes. Thus, edges in $\mathcal{G}^{meta}$ consist of original edges and feature edges. Formally,

$$\mathcal{E}^{meta} = \mathcal{E} \cup \mathcal{E}^f \tag{3}$$

Figure 1 illustrates this transformation. Since every node in $\mathcal{G}^{meta}$ has two types of neighbors, we specify each possible neighborhood.

**1) Graph neighborhood:** nodes from the original graph, $\{\mathcal{N}_v^{\mathcal{G}} = (u \mid (u, v) \in \mathcal{E}\}$,

**2) Feature neighborhood for graph nodes:** feature nodes $\{\mathcal{N}_v^f = (u \mid (u, v) \in \mathcal{E}^f\} \ \forall v \in \mathcal{V}$

**3) Graph neighborhood for features nodes:** graph nodes $\{\mathcal{N}_f^{\mathcal{G}} = (u \mid (u, f) \in \mathcal{E}^f\} \ \forall f \in \mathcal{V}^f$.

### 3.2 Transformation Benefits

In dense attention, each node ingests information from every other node in a single hop, which requires $\mathcal{O}(N^2)$ computations. The proposed transformation enables nodes to exchange information with *non-local* nodes indirectly via feature nodes, thereby avoiding the explicit computation of pairwise attention scores. This transformation preserves the locality biases and induces connections to distant nodes using feature-connectivity biases. Thus, such transformations increase modeling capacity by incorporating long-range interactions to learn inductive biases of both homophilic and heterophilic graphs.

**Connectivity analysis:** We assume $D^{\mathcal{G}}$ to be the average graph node degree of graph nodes $\mathcal{V}$, $D^F$ to be the average no. of features for graph nodes $\mathcal{V}$, and $F^{\mathcal{G}}$ be average no. of nodes per feature nodes $\mathcal{V}^f$. We now show that the transformation drastically increases the connectivity in the following theorem.

**Theorem** 1 (Approximate Connectivity in $\mathcal{G}^{meta}$). *Assuming a tree-like neighborhood expansion and ignoring repeated nodes, the average connectivity of a node in an L-hop neighborhood in an input graph $\mathcal{G}$ grows as $\mathcal{O}((D^{\mathcal{G}})^L)$. Let $\mathcal{G}^{meta}$ be a proposed transformed variant of $\mathcal{G}$. The L-hop neighborhood of same graph node in $\mathcal{G}^{meta}$ grows approximately as $\mathcal{O}((D^{\mathcal{G}})^L + (D^F)^{L/2} * (F^{\mathcal{G}})^{L/2})$.*

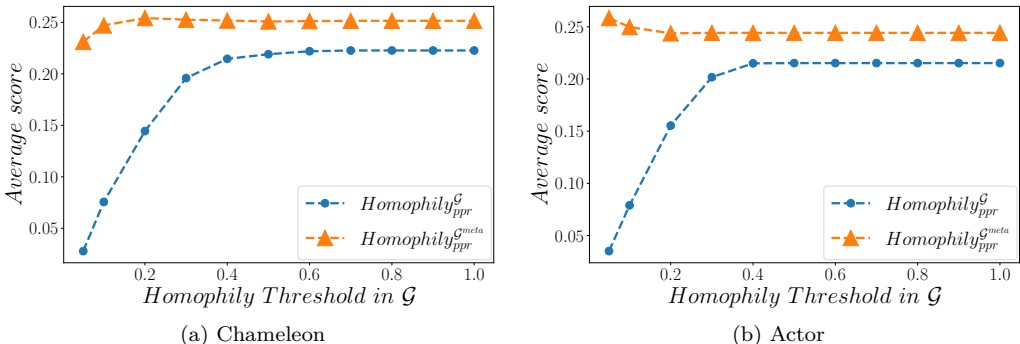

Figure 2: Increase in homophily observed in the transformed graph $\mathcal{G}^{meta}$, while the homophily was low in the original graph $\mathcal{G}$. $K$ is chosen as 50.

**Proof:** See App. B.1. □.

Generally $D^F > D^{\mathcal{G}}$ and $F^{\mathcal{G}} \gg D^{\mathcal{G}}$ in real world graphs. This leads to a significant increase in connectivity. The enhanced connectivity leads to quicker reachability to long-range nodes, facilitating distant nodes to become higher personalized rank nodes(Page et al., 1999) for target nodes.

***Corollary*** 1. $\mathcal{G}^{meta}$ facilitates long-distance nodes in $\mathcal{G}$ to have better personalized page ranks (PPR) from a target node by introducing additional short paths through feature nodes.

**Proof:** See App. B.2.

**Higher homophily:** The increased connectivity due to the proposed transformation leads to higher homophily in top PPR nodes in heterophilic graphs. We demonstrate this empirically on two heterophilic graphs, Actor and Chameleon. We compute the top-$K$ nearest PPR nodes for all graph nodes in $\mathcal{G}$ and $\mathcal{G}^{meta}$ and compute the following homophily score per node. $\forall v \in \mathcal{V}$,

$$Homophily_{ppr}^{\mathcal{G}}(v) = \sum_{i=1}^{i=K} \frac{\mid y(v) = y(ppr_i^{\mathcal{G}}(v)) \mid}{K} \tag{4}$$

where $y(v)$ denotes the label of node $v$, $ppr_i(v)^{\mathcal{G}}$ denotes the $i^{th}$ nearest PPR nodes from node $v$ when computed on graph $\mathcal{G}$. We compute the scores for graph nodes on graph $\mathcal{G}$ and $\mathcal{G}^{meta}$. We find that $Homophily_{ppr}$ increases in $\mathcal{G}^{meta}$ for nodes who had lower $Homophily_{ppr}$ in the original graph. To demonstrate this, we define a homophily threshold to identify nodes whose score is less than the threshold in $\mathcal{G}$. Finally, we compare their average homophily score in $\mathcal{G}$ with the same identified nodes in $\mathcal{G}^{meta}$. Their homophily score increases drastically in $G^{meta}$. We note that during top-$K$ ppr node computation in $\mathcal{G}^{meta}$, we pick only graph nodes, disregarding feature nodes $\mathcal{V}^f$. Figure 2 establishes this phenomenon. The difference reduces once the threshold increases more than 0.5.

Next, we discuss our novel attention mechanism on the transformed graph $\mathcal{G}^{meta}$.

### 3.3 Graph Transformer

The main contribution of this work is to propose features that transform an $N$ dimensional node sub-space into a lower $F$-dimensional feature subspace, enabling attention computation over feature nodes rather than graph nodes where $F = |\mathcal{F}|$. Specifically, we can construct a node-feature incidence matrix, $\mathbf{M} \in \{0,1\}^{N \times F}$ where $\mathbf{M}[i][j] = 1$ when feature $j$ is available in $i^{th}$ node. This projection matrix can reduce the complexity of the core attention operation SOFTMAX$(\mathbf{X}\mathbf{W}_Q(\mathbf{X}\mathbf{W}_K)^T/\sqrt{d})\mathbf{X}\mathbf{W}_V$ with SOFTMAX$((\mathbf{X}\mathbf{W}_Q(\mathbf{X}\mathbf{W}_K)^T/\sqrt{d})\mathbf{M})\mathbf{M}^T\mathbf{X}\mathbf{W}_V$ where $\mathbf{W}_Q, \mathbf{W}_K, \mathbf{W}_V \in \mathbb{R}^{d \times d}$. This projection matrix $\mathbf{M}$ facilitates to project $\mathbf{X}\mathbf{W}_K, \mathbf{X}\mathbf{W}_V \in \mathbb{R}^{N \times d}$ to $\mathbb{R}^{F \times d}$ by multiplying by $\mathbf{M}$, thus reducing the computation to $\mathbb{O}(N*F*d)$ much less than $\mathbb{O}(N*N*d)$ when $F \ll N$. Moreover, unlike virtual token-based methods, which

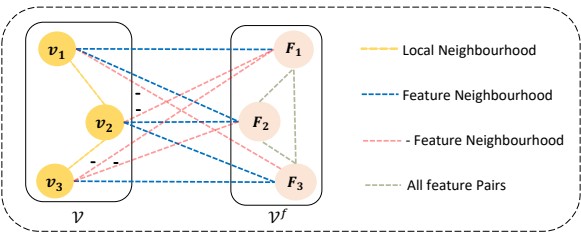

Figure 3: Various attention paths in NEUTAG.

learn the projection matrix $\mathbf{M}$ during training, this approach proposes a fixed, easily derivable projection matrix $\mathbf{M}$. We later discuss the approximation capabilities of this approach in Theorem 2.

NEUTAG architecture constructs the attention graph consisting of 4 types of undirected attention edges as shown in fig 3. These attention paths are applied successively across each layer of NEUTAG. The input to the NEUTAG is the node features $\mathbf{X}$, graph topology $\mathcal{E}$, and position encodings (PE) based on random-walk/Laplacian eigenvalue. The input feature matrix is initialized as

$$\mathbf{H}_{\mathcal{V}}^0 = (\mathbf{X} + \mathbf{PE})\mathbf{W} \tag{5}$$

where $\mathbf{H}_{\mathcal{V}}^0 \in \mathbb{R}^{N \times d}$ is a initial representation of graph nodes $\mathcal{V}$. Similarly, for feature nodes $\mathcal{V}^f$, we initialize their embedding to randomly initialized random vectors.

$$\mathbf{H}_{\mathcal{V}^f}^0[f] = w_f \in \mathcal{R}^d \ \ \forall f \in \mathcal{V}^f \tag{6}$$

Now, given $\mathbf{H}_{\mathcal{V}}^{l-1} \in \mathbb{R}^{N \times d}$, and $\mathbf{H}_{\mathcal{V}^f}^{l-1} \in \mathbb{R}^{F \times d}$, we first compute the following query, key, and value matrices for both graph nodes and feature nodes respectively.

$$\mathbf{Q}_{\mathcal{V}}^l = \mathbf{H}_{\mathcal{V}}^{l-1}\mathbf{W}_1^l, \ \mathbf{K}_{\mathcal{V}}^l = \mathbf{H}_{\mathcal{V}}^{l-1}\mathbf{W}_2^l, \ \mathbf{V}_{\mathcal{V}}^l = \mathbf{H}_{\mathcal{V}}^{l-1}\mathbf{W}_3^l \tag{7}$$

$$\mathbf{Q}_{\mathcal{V}^f}^l = \mathbf{H}_{\mathcal{V}^f}^{l-1}\mathbf{W}_4^l, \ \mathbf{K}_{\mathcal{V}^f}^l = \mathbf{H}_{\mathcal{V}^f}^{l-1}\mathbf{W}_5^l, \ \mathbf{V}_{\mathcal{V}^f}^l = \mathbf{H}_{\mathcal{V}^f}^{l-1}\mathbf{W}_6^l \tag{8}$$

Now we define the following attentions to compute $\mathbf{H}_{\mathcal{V}}^l$ and $\mathbf{H}_{\mathcal{V}^f}^l$. Please note that we provide details using one head for simplicity, but we employ multi-head attention as prevalent in transformers, which entails running attention $H$ times and concatenating these outputs.

**Local neighborhood attention:** Local neighborhood plays a critical role in node classification accuracy and is commonly used in every sparse transformer, e.g. GRAPHGPS, EXPHORMER, GOAT, LARGEGT, and NAGPHORMER. For each target graph node $v \in \mathcal{V}$, the following vector is computed.

$$\mathbf{H}_{\mathcal{V}:local}^l[v] = \sum_{u \in \mathcal{N}_v^{\mathcal{G}}} \frac{\exp(\mathbf{Q}_{\mathcal{V}}^l[v] * \mathbf{K}_{\mathcal{V}}^l[u]/\sqrt{d})}{\sum_{u' \in \mathcal{N}_v^{\mathcal{G}}} \exp(\mathbf{Q}_{\mathcal{V}}^l[v] * \mathbf{K}_{\mathcal{V}}^l[u']/\sqrt{d})} \mathbf{V}_{\mathcal{V}}^l[u] \tag{9}$$

We apply all-pair attention for each feature node $f \in \mathcal{V}^f$ to compute their local representation at layer $l$.

$$\mathbf{H}_{\mathcal{V}^f:local}^l = \text{SOFTMAX}\left(\frac{\mathbf{Q}_{\mathcal{V}^f}^l(\mathbf{K}_{\mathcal{V}^f}^l)^T}{\sqrt{d}}\right)\mathbf{V}_{\mathcal{V}^f}^l \tag{10}$$

**Attention using feature connections:** The next component utilizes graph nodes to feature nodes connections and vice-versa to learn non-local representations.

For graph nodes $\forall v \in \mathcal{V}$, we use the following.

$$\mathbf{H}_{\mathcal{V}:+}^l[v] = \sum_{f \in \mathcal{N}_v^f} \frac{\exp(\mathbf{Q}_{\mathcal{V}}^l[v] * \mathbf{K}_{\mathcal{V}^f}^l[f]/\sqrt{d})}{\sum_{f' \in \mathcal{N}_v^f} \exp(\mathbf{Q}_{\mathcal{V}}^l[v] * \mathbf{K}_{\mathcal{V}^f}^l[f']/\sqrt{d})} \mathbf{V}_{\mathcal{V}^f}^l[f] \tag{11}$$

For feature nodes $\forall f \in \mathcal{V}^f$, we compute the following.

$$\mathbf{H}^l_{\mathcal{V}^f:+}[f] = \sum_{u \in \mathcal{N}^{\mathcal{G}}_f} \frac{\exp(\mathbf{Q}^l_{\mathcal{V}^f}[f] * \mathbf{K}^l_{\mathcal{V}}[u]/\sqrt{d})}{\sum_{u' \in \mathcal{N}^{\mathcal{G}}_f} \exp(\mathbf{Q}^l_{\mathcal{V}^f}[f] * \mathbf{K}^l_{\mathcal{V}}[u']/\sqrt{d})} \mathbf{V}^l_{\mathcal{V}}[u] \tag{12}$$

**Attention using absent feature connections** Since absent features also provide valuable *exclusion* information for node classification, we define negative feature edges as follows.

$$\mathcal{E}^{f-} = \{(v, f) \mid v \in \mathcal{V}, f \in F, \mathbf{X}[v, f] = 0\} \tag{13}$$

Corresponding to negative edges, we define the negative feature neighborhood for graph nodes $V^{\mathcal{G}}$ as $\{\mathcal{N}^-_v = (u \mid (u, v) \in \mathcal{E}^{f-}\}$.

Similar to eq. 9 and 11, we utilize the negative graph connections between graph nodes, and feature node connections are utilized as follows. $\forall v \in \mathcal{V}$,

$$\mathbf{H}^l_{\mathcal{V}:-}[v] = \sum_{f \in \mathcal{N}^-_v} \frac{\exp(\mathbf{Q}^l_{\mathcal{V}}[v] * \mathbf{K}^l_{\mathcal{V}^f}[f]/\sqrt{d})}{\sum_{f' \in \mathcal{N}^-_v} \exp(\mathbf{Q}^l_{\mathcal{V}}[v] * \mathbf{K}^l_{\mathcal{V}^f}[f']/\sqrt{d})} \mathbf{V}^l_{\mathcal{V}^f}[f] \tag{14}$$

Since each graph node has multiple negative features, we sample a fixed number of negative-feature nodes per node using degree-based sampling to improve implementation efficiency.

Finally, these local and non-local representations are merged to learn next-layer representations of graph and feature nodes as follows.

$$\mathbf{H}^l_{\mathcal{V}} = \text{UPDATE}^l_1(\mathbf{H}^{l-1}_{\mathcal{V}}, \text{MLP}^l_1(\mathbf{H}^l_{\mathcal{V}:local} \mid \mathbf{H}^l_{\mathcal{V}:+} \mid \mathbf{H}^l_{\mathcal{V}:-})) \tag{15}$$

$$\mathbf{H}^l_{\mathcal{V}^f} = \text{UPDATE}^l_2(\mathbf{H}^{l-1}_{\mathcal{V}^f}, \text{MLP}^l_2(\mathbf{H}^l_{\mathcal{V}^f:local} \mid \mathbf{H}^l_{\mathcal{V}^f:+})) \tag{16}$$

Here $\text{UPDATE}^l$ can be a neural net-based functions, e.g. MLP or skip-connections.

**Computational Complexity:** Considering all attention components, per-layer complexity of NEUTAG is $\mathcal{O}(Nd^2 + Fd^2 + |\mathcal{E}|d + |\mathcal{E}^f|d + F^2d + Nrd)$ where the first and second terms correspond to linear projections of graph nodes and feature nodes, respectively. The term $|\mathcal{E}|d$ corresponds to the cost of local neighborhood attention, and $|\mathcal{E}^f|d$ corresponds to node-to-feature and feature-to-node attention scaling with node-feature edges. The term $F^2d$ corresponds to feature-feature attention and $Nrd$ to sampled negative feature attention, where $r$ is the number of negative features per node and is small. We note that this complexity is upper-bounded by $\mathcal{O}(NFd)$ in the extreme case where all graph nodes are connected with all feature nodes. In practice, for a large-scale graph where $F \ll N$ yields $\mathcal{O}(NFd) \ll \mathcal{O}(N^2d)$, enabling scalability of NEUTAG to large-scale datasets. For small-scale datasets where $F = \mathcal{O}(N)$, the computational advantage diminishes over dense attention-based graph transformers. However, proposed novel attention paths through feature nodes still enable effective modeling of both homophilic and heterophilic graphs.

We now show that NEUTAG approximates the Positive Orthogonal Random Projections based sparse-transformer PERFORMER (Choromanski et al., 2020), which kernelizes the softmax operation using Mercer's theorem. We formally define the following theorem.

**Theorem** 2. *Under the stated assumptions, NEUTAG can approximate the following PERFORMER self-attention layer applied to the $l^{th}$ layer node representation $\mathbf{H}^l$ of graph $\mathcal{G}$, using at most **4** attention layers:*

$$\mathbf{h}^{l+1}_i = \frac{\phi(\mathbf{W}_Q \mathbf{h}^l_i)^T \sum_{j=1}^{j=N} \phi(\mathbf{W}_K \mathbf{h}^l_j) \otimes (\mathbf{W}_V \mathbf{h}^l_j)}{\phi(\mathbf{W}_Q \mathbf{h}^l_i) \sum_{k=1}^{k=N} \phi(\mathbf{W}_K \mathbf{h}^l_k)} \tag{17}$$

*This holds under the following assumptions: (a) the kernel feature map $\phi$ can be approximated by neural networks, and (b) each graph node is connected to at least one feature node. Here, $\mathbf{h}_i = \mathbf{H}[i] \in \mathbb{R}^d$, $\mathbf{W}_Q, \mathbf{W}_K, \mathbf{W}_V$ are learnable weight matrices, and $\phi : \mathbb{R}^d \to \mathbb{R}^m$ is a low-dimensional feature mapping.*

**Proof:** See App. B.3. □.

Moreover, "at most 4 layer" result in theorem 2 is a theoretical expressivity result, showing NEUTAG can approximate a PERFORMER-style attention. The result establishes that such an approximation is *possible* within 4 layer, but it doesn't imply that fewer layers can't learn an effective representation. Moreover, it also doesn't imply that exactly 4 layers are required in empirical settings, which might depend on dataset characteristics and the nature of the downstream task. Moreover, we analyze NEUTAG's theoretical capabilities to approximate dense-attention in Appendix B.4.

**Extension of Neutag to continuous-valued attributed graphs:** Section C also discusses the extension of NEUTAG on continuous-attributed datasets.

### 3.4 Neutag Mini-Batching

The proposed framework is applicable for **a) Small graphs** by running forward pass on entire graph $\mathcal{G}^{meta}$ and **b) Large-scale graphs** by offline sampling $L$ layer directed sub-graphs from feature nodes $\mathcal{V}^f$ to graph nodes $\mathcal{V}$ from $\mathcal{G}^{meta}$, as these sub-graphs will be common among all batches of graph nodes $\mathcal{V}$ and run NEUTAG on such constructed batches and back-propagate. Algorithm 1 summarizes the creation of a mini-batch for a large-scale graph for NEUTAG training and inference. Specifically, for each feature node $v^f \in \mathcal{V}^f$, graph nodes are sampled in line 4. Corresponding to these graph nodes, the $L$ hop sub-graph is sampled in the original node space $\mathcal{V}$. Please note that data from lines 3-12 can be cached across multiple batches and performed offline. Finally, a batch of nodes is sampled from the input graph, and the corresponding L-HOP subgraph is sampled. Finally, feature edges are added to correspond to these sampled original graph nodes.

## 4 Experiments

### 4.1 Empirical Evaluation

We now evaluate the effectiveness of NEUTAG on node classification tasks across diverse graph datasets and examine its robustness with state-of-the-art graph transformers(GT). We also compare NEUTAG with standard Graph Neural Networks (GNN). Finally, we analyze the importance of feature nodes based on the attention layer via an ablation study. We also analyze the impact of feature sparsity or missing features on NEUTAG's performance in the appendix section E.3.

### 4.2 Datasets

Table 5 in appendix D.1 summarizes the datasets and their statistics. Cora (Sen et al., 2008), CiteSeer (Yang et al., 2016) and OGBN-Arxiv (Hu et al., 2020) are homophilic datasets while Actor (Pei et al., 2020), Chameleon (Rozemberczki et al., 2021), OGBN-Arxiv(year) (Hu et al., 2020) and Snap-Patents (Lim et al., 2021) are heterophilic datasets. Out of these, Snap-Patents is the largest dataset, having 2.92 million nodes and 13.97 million edges. We use 60%, 20%, and 20% train, validation, and test splits on all datasets for all methods, including baselines. More details on datasets and experiment settings, including hyperparameter values, are given in Appendices D.1 and D.2. The codebase is shared at `https://github.com/data-iitd/neutag`.

### 4.3 Baselines

We consider state-of-the-art graph transformers for comparison. We evaluate NEUTAG against GRAPHGPS (Rampášek et al., 2022) and its variant GRAPHGPS-GNN where we remove the GNN component to demonstrate the massive decrease in performance and henceforth dependency on MPNN. Similarly, we evaluate against EXPHORMER (Shirzad et al., 2023) and its variant EXPHORMER-GNN as well as GRIT (Ma et al., 2023), KAA (Fang et al., 2025),GRAPHORMER (Ying et al., 2021), NAGPHORMER (Chen et al., 2022b), GOAT (Kong et al., 2023) and LARGEGT (Dwivedi et al., 2023b).

Moreover, we further evaluate NEUTAG against standard and foundational graph neural networks GRAPH-SAGE (Hamilton et al., 2017), GAT (Veličković et al., 2018), GIN (Xu et al., 2019), LINKX (Lim et al., 2024) and MIXHOP (Abu-El-Haija et al., 2019). MIXHOP solves the over-smoothing in GNN while LINKX is a

---

**Algorithm 1** NEUTAG Mini-batching algorithm

---

**Require:** $\mathcal{G}^{meta} = (\mathcal{V}^{meta} = \mathcal{V} \cup \mathcal{V}^f, \mathcal{E}^{meta} = \mathcal{E} \cup \mathcal{E}^f)$, # of layers $L$, Node feature matrix $\mathbf{X}$
**Ensure:** Sampled mini batch $\mathcal{G}^{meta'}$
1: $\mathcal{V}^{meta'} = \{\}$
2: $\mathcal{E}^{meta'} = \{\}$
   {Sample a $L$ hop subgraph from each feature node. This step can be cached as well as it will remain same across all batches}
3: **for** $v^f \in \mathcal{V}^f$ **do**
4:     $\mathcal{V}^{sampled} \sim 1\text{-HOP}(\mathcal{G} = (\mathcal{V}, \mathcal{E}^f), v^f)$
5:     $(\mathcal{V}^L, \mathcal{E}^L) \sim L\text{-HOP}(\mathcal{G} = (\mathcal{V}, \mathcal{E}), \mathcal{V}^{sampled})$
6:     $\mathcal{V}^{meta'} \leftarrow \mathcal{V}^{meta'} \cup \mathcal{V}^L$
7:     $\mathcal{E}^{meta'} \leftarrow \mathcal{E}^{meta'} \cup \mathcal{E}^L$
8:     **for** $v \in \mathcal{V}^{sampled}$ **do**
9:         $\mathcal{E}^{meta'} \leftarrow \mathcal{E}^{meta'} \cup (v, v^f)$
10:    **end for**
11:    $\mathcal{V}^{meta'} \leftarrow \mathcal{V}^{meta'} \cup v^f$
12: **end for**
    {Sample a batch of original graph nodes and their $L$ hop neighbors}
13: $\mathcal{V}' \sim \mathcal{V}$
14: **for** $v \in \mathcal{V}'$ **do**
15:    $(\mathcal{V}^L, \mathcal{E}^L) \sim L\text{-HOP}(\mathcal{G} = (\mathcal{V}, \mathcal{E}), v)$
16:    $\mathcal{V}^{meta'} \leftarrow \mathcal{V}^{meta'} \cup \mathcal{V}^L$
17:    $\mathcal{E}^{meta'} \leftarrow \mathcal{E}^{meta'} \cup \mathcal{E}^L$
18: **end for**
    {Sample feature nodes and edges for sampled graph nodes to create a mini-batch for forward propagation}

19: **for** $v \in \mathcal{V}^{meta'}$ **do**
20:    **if** $v \in \mathcal{V}$ **then**
21:        $\mathcal{E}^{meta'} \leftarrow \mathcal{E}^{meta'} \cup \{(v, f) \mid f \in \mathcal{V}^f, \mathbf{X}[v, f] = 1\}$
22:    **end if**
23: **end for**
24: **Return** $\mathcal{G}^{meta'} = (\mathcal{V}^{meta'}, \mathcal{E}^{meta'})$

---

Table 1: Comparison of NEUTAG against baseline GT on node classification task

| Method | Cora | CiteSeer | Actor | Chameleon | OGBN-Arxiv | OGBN-Arxiv(Year) | Snap-patents |
|---|---|---|---|---|---|---|---|
| GRAPHGPS | $83.65 \pm 2.67$ | $76.25 \pm 1.34$ | $34.30 \pm 0.45$ | $42.87 \pm 1.88$ | OOM | OOM | OOM |
| GRAPHGPS-GNN | $72.47 \pm 1.87$ | $71.59 \pm 2.43$ | $37.10 \pm 1.11$ | $47.36 \pm 2.22$ | OOM | OOM | OOM |
| EXPHORMER | $86.48 \pm 2.15$ | $75.92 \pm 1.88$ | $35.19 \pm 0.94$ | $45.17 \pm 2.56$ | OOM | OOM | OOM |
| EXPHORMER-GNN | $82.35 \pm 1.75$ | $73.01 \pm 1.20$ | $35.44 \pm 0.86$ | $46.97 \pm 0.95$ | OOM | OOM | OOM |
| GRIT | $82.56 \pm 1.80$ | $76.10 \pm 0.67$ | $35.34 \pm 0.76$ | $48.81 \pm 2.26$ | OOM | OOM | OOM |
| GRAPHORMER | $39.45 \pm 10.66$ | OOM | OOM | $26.89 \pm 7.25$ | OOM | OOM | OOM |
| KAA | $82.16 \pm 1.36$ | $71.83 \pm 1.51$ | $34.88 \pm 0.89$ | $45.44 \pm 5.61$ | OOM | OOM | OOM |
| NAGPHORMER | $86.78 \pm 0.77$ | $74.69 \pm 1.06$ | $33.03 \pm 0.75$ | $59.97 \pm 1.72$ | $67.36 \pm 0.12$ | $48.98 \pm 0.23$ | $61.27 \pm 0.13$ |
| GOAT | $84.93 \pm 0.51$ | $76.75 \pm 1.84$ | $\mathbf{37.98 \pm 1.02}$ | $53.28 \pm 2.48$ | $\mathbf{72.17 \pm 0.09}$ | $50.81 \pm 0.36$ | $55.35 \pm 2.24$ |
| LARGEGT | $83.42 \pm 1.21$ | $70.78 \pm 1.62$ | $37.47 \pm 1.62$ | $57.19 \pm 1.89$ | $67.56 \pm 0.20$ | $53.46 \pm 0.78$ | $\mathbf{63.15 \pm 0.002}$ |
| NEUTAG | $\mathbf{87.67 \pm 1.10}$ | $\mathbf{77.68 \pm 1.90}$ | $36.21 \pm 1.2$ | $\mathbf{65.26 \pm 2.43}$ | $70.63 \pm 0.29$ | $\mathbf{53.96 \pm 0.38}$ | $63.00 \pm 0.22$ |

strong benchmark method for non-homophilic graphs. There exist multiple complementing techniques which enhance GNN performance, e.g., label-propagation (Huang et al., 2021), adaptive channel mixing (Luan et al., 2024b), gradient-gating (Rusch et al., 2023), data-augmentation (Zhao et al., 2022), (Chowdhury et al., 2023) and knowledge-distillation (Hong et al., 2024). Although these techniques can potentially affect GT architectures, study of their effects is beyond the scope of this paper, and we leave it for future work.

For completeness, we also compare NEUTAG with alternative attention formulations in graph transformers, specifically DIFFORMER (Wu et al., 2023a),SGFORMER (Wu et al., 2023b), POLYNORMER (Deng et al.,

2024), and ADVDIFFORMER (Wu et al., 2025). These methods provide equivalent attention formulations that are complementary to sparse graph transformers and can potentially be integrated with NEUTAG and other baselines GOAT, NAGPHORMER, GRAPHGPS, EXPHORMER, and LARGEGT. Moreover, as discussed in the related work section 1.1, there exists a plethora of work focusing on improving positional and structural encodings, tokenization strategies, or other orthogonal design aspects, which are beyond the scope of this study.

## 4.4 Result Analysis

**Comparison with Graph Transformers:** Table 1 presents the node classification accuracy of baselines and the proposed model NEUTAG against 7 diverse datasets. The results clearly demonstrate the strong performance of the proposed model NEUTAG with respect to baselines. As we outlined in the introduction, while GRAPHGPS and EXPHORMER perform well on homophilic datasets, their performance drastically deteriorates after removing the GNN component (-GNN), which improves their performance on heterophilic graphs. Moreover, neither method is scalable for large-scale datasets. The recently proposed graph transformers GOAT, NAGPHORMER, and LARGEGT are scalable via global nodes; their performance is inconsistent across all graphs. E.g., GOAT doesn't perform well on Cora, Chameleon, OGBN-Arxiv(year), and Snap-patents while good on CiteSeer, OGBN-Arxiv, and Actor. LARGEGT is overall worse on small-scale graphs but delivers good performance on large-scale datasets. To further validate the effectiveness of NEUTAG, we extend our comparison with these scalable graph transformers in table 7 in the appendix to 5 additional small but challenging heterophilic datasets as defined in the survey (Luan et al., 2024a). Table 7 clearly demonstrates the strong performance of NEUTAG across these challenging heterophilic graphs. While no model outperforms all baselines across all datasets due to diverse inductive biases, our proposed data-agnostic sparse GT model NEUTAG adapts well to miscellaneous graphs, which is further supported by an analysis of the relative contributions of structural and feature-based attention paths (Appendix E.6). NEUTAG delivers stable performance, consistently ranking best or within 1.5% of the top model across all datasets.

**Comparison with alternate attentions:** Table 3 compares NEUTAG against alternative attention-based graph transformers, including DIFFORMER, SGFORMER, and POLYNORMER. DIFFORMER has two variants: DIFFORMER-s and DIFFORMER-a, where the latter employs a non-linear kernel but fails to scale on medium-sized graphs. Neither variant is able to scale to large graphs like snap-patents. Among these baselines, POLYNORMER performs competitively with NEUTAG. However, both SGFORMER and POLYNORMER require partitioning of large graphs into smaller subgraphs, leading to suboptimal results due to limited information exchange in case of snap-patent, unlike sparse GT NEUTAG, which utilizes information exchange between all nodes through its novel propagation framework. We note that these alternative attention mechanisms can be integrated into NEUTAG and other sparse GT baselines to enhance their performance, which we leave for future work.

**Comparison with Graph Neural Networks:** Table 2 presents the performance of NEUTAG against foundational GNN on 7 datasets. Since information propagation in GNN is limited to a few hops, we clearly see that they are competitive with NEUTAG only on homophilic graphs, Cora, and CiteSeer. Consequently, GNN exhibits worse performance than NEUTAG on Chameleon, OGBN-Arxiv(year), and snap-patents, which require long-range interactions. This clearly establishes the necessity for information propagation from distant nodes for an optimal node classification model. Out of baselines, GRAPHSAGE and MIXHOP are consistent performers. Since MIXHOP utilizes a normalized Laplacian matrix, it doesn't scale to large graphs.

Table 2: Comparison of NEUTAG with GNN on node classification task

| Method | Cora | CiteSeer | Actor | Chameleon | OGBN-Arxiv | OGBN-Arxiv(Year) | Snap-patents |
|---|---|---|---|---|---|---|---|
| GRAPHSAGE | $87.31 \pm 0.96$ | $76.55 \pm 1.78$ | $34.74 \pm 1.20$ | $48.95 \pm 3.16$ | $61.71 \pm 0.79$ | $46.34 \pm 0.25$ | $49.04 \pm 0.03$ |
| GAT | $86.56 \pm 1.13$ | $76.43 \pm 2.55$ | $30.03 \pm 0.67$ | $44.74 \pm 3.29$ | $62.35 \pm 0.20$ | $44.62 \pm 0.52$ | $36.64 \pm 0.53$ |
| GIN | $84.39 \pm 0.65$ | $75.47 \pm 1.28$ | $26.24 \pm 0.52$ | $32.68 \pm 3.68$ | $59.35 \pm 0.30$ | $46.60 \pm 0.29$ | $47.61 \pm 0.12$ |
| MIXHOP | $87.65 \pm 0.20$ | $76.97 \pm 0.99$ | $35.03 \pm 0.53$ | $47.68 \pm 2.89$ | $62.79 \pm 0.39$ | $44.80 \pm 0.17$ | OOM |
| LINKX | $83.14 \pm 1.62$ | $73.72 \pm 0.14$ | $32.78 \pm 0.17$ | $48.20 \pm 3.31$ | $60.39 \pm 0.32$ | $49.00 \pm 0.39$ | $52.71 \pm 0.19$ |
| NEUTAG | $\mathbf{87.67 \pm 1.10}$ | $\mathbf{77.68 \pm 1.90}$ | $\mathbf{36.21 \pm 1.2}$ | $\mathbf{65.26 \pm 2.43}$ | $70.63 \pm 0.29$ | $\mathbf{53.96 \pm 0.38}$ | $\mathbf{63.00 \pm 0.22}$ |

Table 3: Comparison of Neutag against alternate attention formulation based GT

| Method | Cora | CiteSeer | Actor | Chameleon | OGBN-Arxiv | OGBN-Arxiv(year) | Snap-patents | Average |
|---|---|---|---|---|---|---|---|---|
| DIFFormer-s | $87.34 \pm 1.52$ | $\mathbf{77.75 \pm 2.76}$ | $31.20 \pm 0.81$ | $57.41 \pm 2.41$ | $40.45 \pm 1.69$ | $36.74 \pm 0.43$ | OOM | NA |
| DIFFormer-a | $86.01 \pm 2.28$ | $76.70 \pm 1.95$ | $30.79 \pm 1.13$ | $58.07 \pm 1.95$ | OOM | OOM | OOM | NA |
| SGFormer | $86 \pm 1.76$ | $75.83 \pm 2.28$ | $31.03 \pm 2.99$ | $65.77 \pm 1.68$ | $74.51 \pm 0.31$ | $49.14 \pm 0.34$ | $29.44 \pm 0.84$ | 59.03 |
| Polynormer | $87.49 \pm 1.01$ | $75.62 \pm 0.92$ | $\mathbf{37.22 \pm 1.60}$ | $\mathbf{67.63 \pm 1.65}$ | $\mathbf{74.85 \pm 0.15}$ | $52.12 \pm 0.31$ | $31.99 \pm 0.24$ | 60.98 |
| Polynormer (-Gnn) | $71.66 \pm 0.65$ | $70.90 \pm 1.20$ | $38.16 \pm 1.12$ | $49.39 \pm 1.69$ | $65.84 \pm 0.35$ | $43.04 \pm 0.18$ | $31.10 \pm 0.90$ | 52.87 |
| AdvDIFFormer | $79.08 \pm 1.30$ | $69.58 \pm 1.95$ | $33.30 \pm 0.89$ | $50.48 \pm 5.13$ | $66.83 \pm 0.13$ | $39.26 \pm 0.58$ | OOM | NA |
| Neutag | $\mathbf{87.67 \pm 1.10}$ | $77.68 \pm 1.90$ | $36.21 \pm 1.2$ | $65.26 \pm 2.43$ | $70.63 \pm 0.29$ | $\mathbf{53.96 \pm 0.38}$ | $\mathbf{63.00 \pm 0.22}$ | $\mathbf{64.91}$ |

Linkx designed for heterophilic graphs is competitive on Chameleon and outperforms the rest of the Gnn on large-scale snap-patent.

**Scalability:** As shown in Table 1, dense-attention-based GT models such as GraphGPS, Exphormer, and Graphormer do not scale to large datasets such as OGBN-Arxiv, OGBN-Arxiv(year), and Snap-patents due to their computational and memory requirements. In contrast, scalable graph transformers such as NagPhormer, Goat, and LargeGT, along with our proposed method Neutag, can handle large-scale graphs. We provide additional analysis of training time and model size in Tables 11 and 12 (Appendix E.4). While Neutag is not the fastest or most lightweight among scalable methods, it achieves consistent performance across diverse settings, including both small- and large-scale datasets, and homophilic and heterophilic graphs, whereas none of these baselines do.

**Ablation Study:** We refer readers to the Appendix E.1 and E.7 to analyze the impact of various attention components and hyper-parameters in Neutag. Additionally, we conduct studies to understand the impact of missing features on the performance of Neutag in Appendix E.3.

## 5 Limitations

The major limitation of our work is that the proposed Neutag is not applicable to non-attributed graphs.

Moreover, the proposed method is specifically designed for node classification tasks in both small and large-scale graphs. In contrast, graph classification tasks involve small graphs, e.g., pattern, cluster, zinc, peptides-func, peptides-struct (Dwivedi et al., 2023a; 2022b), having around 100 nodes on average. In such cases, dense GT has shown to perform well without any computational challenges (Rampášek et al., 2022; Shirzad et al., 2023; Ying et al., 2021; Ma et al., 2023).

## 6 Conclusion

We introduced Neutag, a novel sparse graph transformer that unifies structural and feature information within a single attention mechanism. Unlike prior approaches relying on separate GNN components or virtual nodes, Neutag leverages features as global nodes, enabling long-range connectivity on attributed graphs. We further provide theoretical analysis on Neutag's capabilities. Finally, experiments on seven real-world datasets demonstrate that Neutag achieves competitive and consistent performance across diverse graph types, underlining its generality.

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

## A    Preliminaries

**Definition** 1 (Graph). *A graph is defined as $\mathcal{G} = (\mathcal{V}, \mathcal{E}, \mathbf{X})$ over node and edge sets $\mathcal{V}$ and $\mathcal{E} = \{(u,v) \mid u, v \in \mathcal{V}\}$ respectively where $|\mathcal{V}| = N$ and $|\mathcal{E}| = M$. Edge set is also represented using an adjacency matrix $\mathcal{A} \in \{0,1\}^{N \times N}$. $\mathbf{X} \in \{0,1\}^{N \times F}$ is a node feature matrix where $\mathcal{F} = \bigcup_{v \in \mathcal{V}} \mathcal{F}_v$ is the set of all features in graph $\mathcal{G}$. $\mathcal{F}_v$ is a feature set at node $v$ and $|\mathcal{F}| = F$.*

**Problem** 1 (Graph transformer for node classification).

**Input:** *Given a graph $\mathcal{G}$ (Def. 1), let $Y : \mathcal{V} \rightarrow \mathbb{R}$ be a hidden function that maps a node to a real number. $Y(v)$ is known to us only for the subset $\mathcal{V}_l \subset \mathcal{V}$ and may model some downstream tasks such as node classification or link prediction.*

**Goal:** *Learn parameters $\Theta$ of a Transformer based graph neural network, denoted as $GT_\Theta$, that predicts $Y(v)$, $\forall v \in \mathcal{V}_l$ accurately.*

We now introduce preliminaries of Graph Neural Networks, Transformers, and Graph Transformers for node classification tasks.

### A.1    Graph

Gnn also known as message-passing neural networks, as each node exchanges messages from its neighbors to compute its representation. These representations are utilized in downstream tasks such as node classification and link prediction. Though there exists more specialized Gnn for the link-prediction task that utilizes link-based features instead of node-level, those are out of scope in this work. State-of-the-art Gnn (Xu et al., 2019; Veličković et al., 2018; Hamilton et al., 2017) for node classification tasks follows the following framework. Assuming $\mathbf{x}_v \in \mathcal{R}^{|F|}$ as feature vector for node $v$, $0^{th}$ layer embedding is defined as:

$$\mathbf{h}_v^0 = \mathbf{x}_v \ \forall v \in \mathcal{V} \tag{18}$$

Next, $l^{th}$ layer representation is computed using nodes' neighbourhood $\mathcal{N}_v = \{u \mid (u,v) \in \mathcal{E}\} \ \forall \ v \in \mathcal{V}$ as follows.

$$\mathbf{m}_v^l = \text{MSG}(\mathbf{h}_u^{l-1}, \mathbf{h}_v^{l-1}) \forall u \in \mathcal{N}_v \tag{19}$$

Messages are computed from each neighbor using the previous layer information. This information is then aggregated at each node as follows.

$$\overline{\mathbf{m}}_v = \text{AGGREGATE}^l(\{\!\!\{ \mathbf{m}_v^l(u), \forall u \in \mathcal{N}_v \}\!\!\}) \tag{20}$$

| Symbol | Meaning |
|---|---|
| $\mathcal{G}$ | Input graph |
| $\mathcal{V}$ | Set of nodes in $\mathcal{G}$ |
| $\mathcal{E}$ | Set of edges in $\mathcal{G}$ |
| $\mathbf{X}$ | Node feature matrix |
| $\mathcal{F}$ | Set of features in $\mathcal{G}$ |
| $\mathcal{G}^{meta}$ | Transformed input graph |
| $\mathcal{V}^f$ | Set of features as virtual node in $\mathcal{G}^{meta}$ |
| $\mathcal{E}^f$ | Set of edges between nodes and respective features $\mathcal{G}^{meta}$ |
| $\mathcal{E}^{f-}$ | Set of edges between nodes and absent features nodes in $\mathcal{G}^{meta}$ |
| $\mathcal{V}^{meta}$ | Set of nodes in $\mathcal{G}^{meta}$ |
| $\mathcal{E}^{meta}$ | Set of edges in $\mathcal{G}^{meta}$ |
| $D^{\mathcal{G}}$ | Average node degree excluding feature nodes in $\mathcal{G}^{meta}$ |
| $D^F$ | Average node degree excluding graph nodes in $\mathcal{G}^{meta}$ |
| $y(v)$ | Label of node $v$ |
| $ppr_i^{\mathcal{G}}(v)$ | $i^{th}$ nearest node from node $v$ sorted using personalized page rank score |
| $\mathbf{M}$ | Projection matrix |
| $\mathbf{h}_i^l$ | Embedding of node $i$ at $l^{th}$ layer |
| $\mathbf{H}_{\mathcal{V}}^l$ | Embedding matrix of graph nodes $\mathcal{V}$ at layer $l$ |
| $\mathbf{H}_{\mathcal{V}^f}^l$ | Embedding matrix of feature nodes $\mathcal{V}^f$ at layer $l$ |
| $\mathbf{W}$ | Learnable weight matrices |
| $\mathbf{Q}_{\mathcal{V}}^l$ | Query matrix at layer $l$ for graph nodes $\mathcal{V}$ |
| $\mathbf{K}_{\mathcal{V}}^l$ | Key matrix at layer $l$ for graph nodes $\mathcal{V}$ |
| $\mathbf{V}_{\mathcal{V}}^l$ | Value matrix at layer $l$ for graph nodes $\mathcal{V}$ |
| $\mathbf{Q}_{\mathcal{V}^f}^l$ | Query matrix at layer $l$ for feature nodes $\mathcal{V}^f$ |
| $\mathbf{K}_{\mathcal{V}^f}^l$ | Key matrix at layer $l$ for feature nodes $\mathcal{V}^f$ |
| $\mathbf{V}_{\mathcal{V}^f}^l$ | Value matrix at layer $l$ for feature nodes $\mathcal{V}^f$ |
| $\mathcal{N}_v^{\mathcal{G}}$ | Set of neighbors excluding feature nodes of node $v$ in $\mathcal{G}^{meta}$ |
| $\mathcal{N}_v^f$ | Set of neighbors consisting of only feature nodes of node $v$ in $\mathcal{G}^{meta}$ |
| $\mathcal{N}_f^{\mathcal{G}}$ | Set of neighbors excluding feature nodes of a feature node $f$ in $\mathcal{G}^{meta}$ |

Table 4: Notations and their definition

$\{\!\{\ldots\}\!\}$ is a multi-set as the same message can arrive from multiple neighbors. Multi-set allows multiple instances of the same element achieving improved expressivity, highlighted in (Xu et al., 2019). Finally, the aggregated message and previous layer $l-1$ representation are combined to compute the $l^{th}$ layer representation as follows.

$$\mathbf{h}_v^l = \text{COMBINE}(\mathbf{h}_v^{l-1}, \overline{\mathbf{m}}_v) \tag{21}$$

where MSG, AGGREGATE and COMBINE are non-neural functions like SUM, AVERAGE or MAX-POOL or neural networks based learnable functions like *mlp*, *attention* (Vaswani et al., 2017) and *recurrent neural networks* eg. GRU (Dey & Salem, 2017). To achieve $L$ hop deeper GNN, equations 19, 20 and 21 are applied $L$ times successively to compute $\mathbf{h}_v^L$. This representation is utilized for node classification tasks. GNN are limited in modeling long-range dependencies as increasing the number of layers leads to *over-smoothing*(Oono & Suzuki, 2020; Chen et al., 2020) where node embeddings become approximately similar at every node. Graph Transformers solves this by introducing the mechanism of each node attending to all other nodes as follows.

### A.2 Transformers

First, we define the transformer neural nets, the key components of graph transformers. Given a graph $\mathcal{G} = (\mathcal{V}, \mathcal{E}, \mathbf{X})$ and ignoring topological connections $\mathcal{E}$, contextualized node representations $\mathbf{H}^L$ are computed using self-attention. The representations are first initialized using node features.

$$\mathbf{H}^0 = \mathbf{X}\mathbf{W} \tag{22}$$

where $\mathbf{W}$ is the trainable weight matrix. Next, below equations 23 and 24 are repeated for $l \in [1 \ldots L]$ as follows.

$$\mathbf{H}^l = \left\|_{h=1}^{h=H} \text{SOFTMAX} \left( \frac{(\mathbf{H}^{l-1}\mathbf{W}_Q^h)(\mathbf{H}^{l-1}\mathbf{W}_K^h)^T}{\sqrt{d}} \right) \mathbf{H}^{l-1}\mathbf{W}_V^h \tag{23}$$

$$\mathbf{H}^l = \text{NORM}(\mathbf{H}^{l-1} + \text{FFN}(\mathbf{H}^l)) \tag{24}$$

where NORM is either batch-norm (Ioffe & Szegedy, 2015) or layer-norm (Ba et al., 2016), FFN is feed-forward neural network. $\mathbf{W}_Q$, $\mathbf{W}_K$ and $\mathbf{W}_V$ are projection matrix $\in \mathcal{R}^{d \times d_k}$. $H$ is a number of heads, and the transformer concatenates these multiple heads, facilitating diverse attention coefficients. This is also known as *Multi-head Attention*. This architecture uses skip-connections and activation norm strategies required in deeper neural networks (He et al., 2016).

### A.3 Graph Transformers (GT)

Now, we discuss graph transformers, which incorporate graph topology $\mathcal{E}$ into the attention mechanism. Foremost, node attributes are combined with position encodings, e.g., Random walk-based encoding (Dwivedi et al., 2022a) and Laplacian eigenvalues (Dwivedi et al., 2023a) denoted as $\mathbf{PE}$ matrix.

$$\mathbf{H}^0 = (\mathbf{X} + \mathbf{PE})\mathbf{W} \tag{25}$$

Next, assuming we have node presentation for $l - 1$ layer, it is passed to the GNN layer along with the transformer layer to compute $l^{th}$ layer representation as follows.

$$\mathbf{H}_{gnn}^l = \text{GNN}(\mathbf{H}^{l-1}, \mathcal{E}) \tag{26}$$

$$\mathbf{H}_T^l = \text{TRANSFORMER}(\mathbf{H}^{l-1}, \mathcal{E}) \tag{27}$$

where GNN is any graph neural network described earlier. TRANSFORMER layer can be defined as in current literature (Rampášek et al., 2022; Shirzad et al., 2023; Kong et al., 2023; Dwivedi et al., 2023b). Finally, the representation computed using GNN and TRANSFORMER are combined to learn $l^{th}$ layer representation.

$$\mathbf{H}^l = \text{FFN}(\mathbf{H}_{gnn}^l, \mathbf{H}_T^l) \tag{28}$$

## B Proof of Theorems

### B.1 Connectivity Analysis of $\mathcal{G}^{meta}$: Proof of theorem 1

**Proof:** First, we define the number of 1-hop neighbors of the graph and feature nodes where $\#Nbrs(\mathcal{V}, l)$ signifies the order of $l$-hop neighbors of graph nodes $\mathcal{V}$ and $\#Nbrs(\mathcal{V}^f, l)$ is the order of number of $l$-hop neighbors of feature nodes $\mathcal{V}^f$.

$$\#Nbrs(\mathcal{V}, 1) = O(D^\mathcal{G} + D^F), \quad \#Nbrs(\mathcal{V}^f, 1) = O(F^\mathcal{G}) \tag{29}$$

As each $D^F$ feature node will connect to graph nodes connected to it, and each graph node $F^\mathcal{G}$ and $D^\mathcal{G}$ will be connected to its neighbors and feature nodes, applying this for 2 hops,

$$\#Nbrs(\mathcal{V}, 2) = O(D^\mathcal{G} * (D^\mathcal{G} + D^F) + D^F * F^\mathcal{G}),$$
$$\#Nbrs(\mathcal{V}^f, 2) = O(F^\mathcal{G} * (D^\mathcal{G} + D^F)) \tag{30}$$

$$\#Nbrs(\mathcal{V}, 3) = O(D^{\mathcal{G}} * (D^{\mathcal{G}} * (D^{\mathcal{G}} + D^F) + D^F * F^{\mathcal{G}}) + D^F * F^{\mathcal{G}} * (D^{\mathcal{G}} + D^F)) \tag{31}$$

$$\begin{aligned}\#Nbrs(\mathcal{V}, 4) =& O(D^{\mathcal{G}} * (D^{\mathcal{G}} * (D^{\mathcal{G}} * (D^{\mathcal{G}} + D^F) + D^F * F^{\mathcal{G}}) + D^F * F^{\mathcal{G}} * (D^{\mathcal{G}} + D^F)) \\ &+ D^F * F^{\mathcal{G}} * (D^{\mathcal{G}} * (D^{\mathcal{G}} + D^F) + D^F * F^{\mathcal{G}}))\end{aligned} \tag{32}$$

While the closed form for $L$-hop neighbor is not feasible, we re-write $\#Nbrs(\mathcal{V}, 4)$ using its recursive nature,

$$\begin{aligned}\#Nbrs(\mathcal{V}, 4) =& O(D^{\mathcal{G}} * (D^{\mathcal{G}} * \#Nbrs(\mathcal{V}, 2)) + D^F * F^{\mathcal{G}} D^{\mathcal{G}} + (D^F)^2 F^{\mathcal{G}} \\ &+ D^F * F^{\mathcal{G}} * \#Nbrs(\mathcal{V}, 2) + D^F * F^{\mathcal{G}})\end{aligned} \tag{33}$$

where $\#Nbrs(\mathcal{V}, 2)$ contains $D^F * F^{\mathcal{G}}$. Thus, we see that $D^{\mathcal{G}}$ grows with the power of the number of hops $L$, and $D^F * F^{\mathcal{G}}$ multiplies after every other hop. Consequently, we approximate $Nbrs(\mathcal{V}, L)$ as

$$\#Nbrs(\mathcal{V}, L) \approx O((D^{\mathcal{G}})^L + (D^F)^{L/2} * (F^{\mathcal{G}})^{L/2}) \tag{34}$$

$$\square.$$

## B.2 Proof of corollary 1

We can see this via the personalized page-rank ($PPR$) equation for a target node $v$ defined as

$$\pi(v) = (1 - \alpha)\tilde{\mathcal{A}}\pi(v) + \alpha\mathbf{i}_v \tag{35}$$

where $\tilde{\mathcal{A}}$ is normalized with added self-loops, $\mathbf{i}_v$ is an indicator vector with indices except corresponding to node $v$ are filled with 0 and $\alpha \in (0, 1)$ is teleportation probability facilitating transportation to target node $v$ in restart. Under conditions defined in PAGERANK-NIBBLE (Andersen et al., 2006), this equation equation can be expanded as,

$$\pi(v) = \alpha * \mathbf{i}_v(I - (1 - \alpha)\tilde{\mathcal{A}})^{-1} = \alpha * \sum_{r=0}^{i=\infty} (1 - \alpha)^r \tilde{\mathcal{A}}^r \mathbf{i}_v \tag{36}$$

I is an identity matrix the same size as $\tilde{\mathcal{A}}$. As seen in the equation, the nearest nodes have a higher weightage of $(1 - \alpha)$, which decays exponentially $(1 - \alpha)^r$ with longer hops, where $r$ is the shortest distance from the target node. The *feature-as-node* transformation used to construct $\mathcal{G}^{meta}$ introduces 2-hop paths for a pair of nodes $(u, v)$ sharing at-least a single feature $f$ as $v \to f \to u$ in $\mathcal{G}^{meta}$. Thus, the proposed transformation adds shorter paths between nodes that were originally farther in the $\mathcal{G}$, thereby improving their relative personalized page ranks. $\square.$

## B.3 Neutag is an Approximation of a Sparse Transformer Performer: Proof of theorem 2

We use a similar strategy outlined in (Cai et al., 2023), which showed that global nodes can approximate PERFORMER. First, we rewrite the equation 17 as follows to simplify the analysis.

$$\mathbf{h}_i^{l+1} = \frac{\phi_1(\mathbf{h}_i^l)^T \sum_{j=1}^{j=N} \phi_2(\mathbf{h}_j^l) \otimes \phi_3(\mathbf{h}_j^l)}{\phi_1(\mathbf{h}_i^l)^T \sum_{o=1}^{o=N} \phi_2(\mathbf{h}_o^l)} \tag{37}$$

where we define $\phi_1(\mathbf{h}) = \phi(\mathbf{W}_Q\mathbf{h})$, $\phi_2(\mathbf{h}) = \phi(\mathbf{W}_K\mathbf{h})$, and $\phi_3(\mathbf{h}) = \mathbf{W}_V\mathbf{h}$. This allows us to use MLPs' universal approximation capability in the attention and aggregation equations. Intuitively, feature nodes can facilitate approximation of both global summations $\sum_{j=1}^{j=N} \phi_2(\mathbf{h}_j) \otimes \phi_3(\mathbf{h}_j)$ and $\sum_{o=1}^{o=N} \phi_2(\mathbf{h}_o)$ because each graph node is connected to at least 1 feature node.

To prove theorem 2, we assume that in 15, $\mathbf{H}_{\mathcal{V}:local}$ is ignored by UPDATE$_1^l$ as local connectivity of graph nodes is not required to approximate PERFORMER attention. We further subdivide layer $l$ of NEUTAG into $(l1, l2, \ldots)$ for the purpose of this analysis.

Now, at layer $l$, we are given $\mathbf{h}_v \forall v \in \mathcal{V}$, For feature nodes $f \in \mathcal{V}^f$, let the representation include a one-hot identity vector as $\mathbf{h}_f = [\ldots, \mathbb{I}_f] \forall$ where $\mathbb{I}_f$ is a one-hot indicator vector with all zeros except $f^{th}$ index, which

is equal to 1. This indicator vector is preserved from layer $l = 0$ and facilitates computation of feature degree $d_v^F$ for every graph node $v$.

Next, using equations 11 and 16 and by learning equal attention coefficients for all present feature nodes, each graph node $v \in \mathcal{V}$ approximates either of the two representations as follows.

**1)** $[\phi_2(\mathbf{h}_v^l), (\phi_2(\mathbf{h}_v^l) \otimes \phi_3(\mathbf{h}_v^l))_{\text{flattened}}, d_v^F]$  or **2)** $[\phi_2(\mathbf{h}_v^l)/d_v^F, (\phi_2(\mathbf{h}_v^l) \otimes \phi_3(\mathbf{h}_v^l))_{\text{flattened}}/d_v^F]$

where flattened signifies flattening of the matrix to a vector. This corresponds to sub-layer $l1$.

At sub-layer $l2$, each feature node $f \in \mathcal{V}^f$ using equation 12, learn attention coefficients $\frac{1}{d_v^F}$ for all $v \in \mathcal{N}_f^{\mathcal{G}}$ and computes vector

$$\left[ \sum_{v \in \mathcal{N}_f^{\mathcal{G}}} \phi_2(\mathbf{h}_v^l)/d_v^F, \quad \sum_{v \in \mathcal{N}_f^{\mathcal{G}}} (\phi_2(\mathbf{h}_v^l) \otimes \phi_3(\mathbf{h}_v^l))_{\text{flattened}}/d_v^F, \quad \mathbb{I}_f \right].$$

The one-hot indicator vector $\mathbb{I}_f$ is required to ensure a distinct representation for each feature node and to allow for feature-degree statistics at graph nodes in consecutive layers. A similar representation can be learned in Case 2 using equal attention coefficients.

Finally at sub-layer $l3$, feature nodes can spread these partial statistics to each graph node using both positive and negative attention paths and consecutively graph nodes can aggregate partial statistics into global $[\sum_{v=1}^{v=N} \phi_2(\mathbf{h}_v^l), \sum_{v=1}^{v=N} (\phi_2(\mathbf{h}_v^l) \otimes \phi_3(\mathbf{h}_v^l))_{flattened}]$ using equations 11,14 and eq. 15 and approximate the required PERFORMER layer 37 using eq. 15 in this final aggregation step.

Alternatively, at sub-layer $l3$, all-pair feature-to-feature attention path facilitates each feature node $f \in \mathcal{V}^f$ to learn equal attention weight of $\frac{1}{F}$ and aggregate global sums $[\sum_{v=1}^{v=N} \phi_2(\mathbf{h}_v^l), \sum_{v=1}^{v=N} (\phi_2(\mathbf{h}_v^l) \otimes \phi_3(\mathbf{h}_v^l))_{flattened}, \mathbb{I}_f]$ using eq. 10 and 16. At sub-layer $l4$, each graph node $v \in \mathcal{V}$ retrieves these global sums using eq. 11 and compute the required performer attention layer 37 using eq. 15.

Either of these two approximation pathways is sufficient to approximate the PERFORMER layer. Consequently, the number of parameters required in each layer of NEUTAG to approximate a PERFORMER layer remains constant with respect to the number of nodes $N$. $\square$.

## B.4 Universal Approximation Analysis of Neutag

Dense transformers have been proven universal approximations of sequence-to-sequence permutation equivariant functions (Yun et al., 2020). The same work further proves that transformers are universal approximators of all sequence-to-sequence functions by including position encoding. Further SAN (Kreuzer et al., 2021a) proves that since a graph can be constructed as a sequence of edges or nodes, dense attention-based graph transformers are universal approximates of such sequences within a bound, inducing higher expressivity than 1-Weisfeiler Lehman (WL) isomorphism test. Since NEUTAG doesn't utilize all $\mathcal{O}(N^2)$ connections, analyzing its universal approximation capabilities is important. Formally,

**Theorem** 3. *Under the stated assumptions, for a given input graph $\mathcal{G}$, its transformed graph $\mathcal{G}^{meta}$, and node representation matrix $\mathbf{X} \in \mathbb{R}^{N \times d}$, there exists a NEUTAG attention layer that can approximate the following all-pair self-attention operation as a permutation-equivariant universal approximator, using $\mathcal{O}(N^d)$ parameters and $\mathcal{O}(1)$ layers:*

$$\mathbf{H} = \text{SOFTMAX}\left( \frac{(\mathbf{H}^l \mathbf{W}_Q)(\mathbf{H}^l \mathbf{W}_K)^T}{\sqrt{d}} \right) \mathbf{H}^l \mathbf{W}_V \tag{38}$$

*This holds under the following assumptions: (i) each graph node $v \in \mathcal{V}$ is connected to at least one feature node $f \in \mathcal{V}^f$ in $\mathcal{G}^{meta}$, and (ii) the model has sufficient capacity $\mathcal{O}(N^d)$.*

**Proof:** Similar to (Cai et al., 2023), we first relate NEUTAG to DEEPSETS (Zaheer et al., 2017), which is a universal approximator of sequence-to-sequence permutation equivariant functions. Formally, we define the following lemma.

**Definition** 2 (DeepSets (Zaheer et al., 2017)). *Each layer of DEEPSETS is defined as follows.*

$$\mathbf{H}^{l+1} = \sigma(\mathbf{H}^l \mathbf{W}_1 + \frac{1}{N} \mathbf{1}\mathbf{1}^T \mathbf{H}^l \mathbf{W}_2) \tag{39}$$

where $\sigma$ is a non-linearity activation function, $\mathbf{H}^l$ is output of previous layer, $\mathbf{1} = [1, 1, \ldots]^T$ is $N$ dimensional vector and $\mathbf{W}_1$, $\mathbf{W}_2$ are learnable weight matrices.

$\frac{1}{N}\mathbf{1}\mathbf{1}^T$ calculates the average of transformed inputs $\mathbf{H}^l\mathbf{W}_2$, producing a permutation-equivariant function. ,which is added to $\mathbf{H}^l\mathbf{W}_1$. This function can easily be verified as permutation-equivariant, since node reordering only permutes the output. Now, we formally write the following lemma of (Segol & Lipman, 2019).

**Lemma B.1** ((Segol & Lipman, 2019)). *DEEPSETS with $\mathcal{O}(1)$ layers and $\mathcal{O}(N^d)$ parameters per layer is a universal approximator for permutation equivariant sequence to sequence functions.*

Thus, DEEPSETS can approximate the self-attention operation in eq. 38, which is a permutation equivalent function. Similar to the proof of theorem 2, both operations in DEEPSETS, **a)** calculating the average of node embeddings and **b)** adding the calculated average to the node representation and applying $\sigma$ can be simulated by NEUTAG leveraging the fact that each graph node is connected to at least one feature node. And since DEEPSETS can approximate equation 38, the result follows. $\qquad\square$

Consequently, while the theorem shows that the proposed transformation is theoretically capable of approximating a dense self-attention layer, this does not imply that such an approximation is achievable with a realistic number of model parameters, especially for large-scale graphs.

**Practical scope and limitations:** The above result shows that NEUTAG can approximate a dense self-attention layer under the stated assumptions. Though this does not imply exact equivalence in practical settings, particularly, the construction in the proof requires $\mathcal{O}(N^d)$ parameters, which is impractical for real-world graphs. Therefore, this result primarily serves as theoretical grounding for the proposed attention mechanism.

## C  Extension of Neutag to real-valued attributed graphs

Given a real-valued attributed graph $\mathcal{G} = (\mathcal{V}, \mathcal{E}, \mathbf{X}^{real})$, we aim to convert it to $\mathcal{G} = (\mathcal{V}, \mathcal{E}, \mathbf{X}^{binary})$ where $\mathbf{X}^{real} \in \mathbb{R}^{N \times d}$ and $\mathbf{X}^{binary} \in \{0,1\}^{N \times d'}$. To achieve this, we utilize the theory of knowledge distillation (Hinton et al., 2015). We first train a teacher model $\theta_{teacher}$, a 2-layer MLP with input $\mathbf{X}^{real}$, on the node classification task. Please note that graph topology is not used for this training. For the student model, we adopt the $k-sparse$ encoder (Makhzani & Frey, 2013).

$$\Gamma = \mathrm{supp}_k(\mathbf{X}^{real}\mathbf{W}) \tag{40}$$

$$\mathbf{X}_\Gamma^{binary} = 1, \quad \mathbf{X}_{\Gamma^\complement}^{binary} = 0 \tag{41}$$

Here $\mathbf{W} \in \mathbb{R}^{d \times d'}$, $\mathrm{supp}_k$ selects the $k$ indices corresponding to $k$ largest activation. Finally, we set those indices to 1 and the rest to 0 in $\mathbf{X}^{binary}$. Finally, we train the student network using the following knowledge-distillation loss.

$$\mathbf{L}^{teacher} = \mathrm{SOFTMAX}(\mathrm{MLP}_{\theta teacher}(\mathbf{X}^{real})) \tag{42}$$

$$\mathbf{L}^{student} = \mathrm{SOFTMAX}(\mathrm{MLP}_{\theta student}(\mathbf{X}^{student})) \tag{43}$$

$$\mathcal{L} = (1-\lambda)\mathrm{KL}(\mathbf{L}^{teacher}, \mathbf{L}^{student}) + \lambda\mathrm{CE}(\mathbf{Y}, \mathbf{L}^{student}) \tag{44}$$

where $\mathbf{L}^{teacher}$ and $\mathbf{L}^{student}$ are class probabilities from teacher and student models respectively. KL is a KL-divergence loss, and CE is a cross-entropy loss for the classification task. $\mathbf{Y}$ is a ground truth vector. $\lambda$ is a hyper-parameter to adjust the weightage between KL divergence loss and cross-entropy loss. We note that $\theta^{teacher}$ is fixed from the teacher model, and $\mathcal{L}$ is used to train the $\theta^{student}$ and $\mathbf{W}$ matrix from eq. 40.

We use $\lambda = 0.1$. The transformed $\mathbf{X}^{binary}$ vectors can achieve classification accuracy within $\sim 10-15\%$ margin compared to $\mathbf{X}^{real}$ vector on the classification task using a MLP without graph topology as input. Specifically, in OBGN-Arxiv, $\mathbf{X}^{real}$ achieves $53\%$ accuracy while $\mathbf{X}^{binary}$ achieves $47\%$. In the case of OGBN-Arxiv(year), it is $33\%$ vs $29\%$.

## D  Additional Experimental Details

### D.1  Datasets

Cora (Sen et al., 2008) and CiteSeer(Yang et al., 2016) are co-citation graphs where nodes are papers, and their features are bag-of-words of text. The task is to predict the research category of the node. Actor (Pei et al., 2020) is a co-occurrence graph of actors on the same wiki page. Node attributes are bag-of-words from the actor's Wikipedia page, and their labels are actor categories. Chameleon (Rozemberczki et al., 2021) is a graph of hyperlinks between English wiki pages, attributes are nouns, and the label is the binned average monthly traffic on the page. Snap-patents (Lim et al., 2021) is a large-scale co-citation graph of U.S. utility patents where attributes are patent metadata and class label is the time at which the patent was granted, binned in 5 classes. OGBN-Arxiv (Hu et al., 2020) is also a co-citation network where features are 128-dimension embeddings of title and abstract, and the label is the research category. OGBN-Arxiv(Year) is the same graph, but the label is the year of publication, and it is a non-homophilic graph. $\mathbf{H}_{edge}$ (Zhu et al., 2020) in table 5 denotes the edge homophily of a graph.

Table 5: Dataset statistics

| Dataset | # Nodes | # Edges | # Features | #Labels | $\mathbf{H}_{edge}$ | Avg. features per node |
|---|---|---|---|---|---|---|
| Cora | 2708 | 10556 | 1433 | 7 | 0.81 | $\approx 18$ |
| CiteSeer | 3327 | 9104 | 3703 | 6 | 0.74 | $\approx 32$ |
| Actor | 34493 | 495924 | 8415 | 5 | 0.22 | $\approx 5$ |
| Chameleon | 7600 | 33544 | 931 | 5 | 0.23 | $\approx 13$ |
| OGBN-Arxiv | 169343 | 1166243 | 128 | 40 | 0.81 | 128 |
| OGBN-Arxiv(year) | 169343 | 1166243 | 128 | 5 | 0.22 | 128 |
| Snap-Patents | 2923922 | 13975788 | 269 | 5 | 0.07 | 5 |

### D.2  Hardware Details

We have performed experiments on an Intel Xeon 6248 processor with a Tesla V-100 GPU with 32GB GPU memory and Ubuntu 18.04. Train, validate, and test data split of 60%, 20%, and 20%, which are generated randomly for every run. We perform 5 runs of every experiment to report the mean and standard deviation. We use $4 - 6$ layer NEUTAG for small graphs and 2 layer for large graphs. We use Adam optimizer to train the model using a learning rate of 0.00001 and choose the best model based on validation loss. For all methods, including baselines and NEUTAG, we apply laplacian position encodings for small-scale datasets, and node2vec based position encoding in the snap-patent dataset, as laplacian position encoding calculation is computationally infeasible at million-scale datasets, as proposed in GOAT, for large-scale datasets. These are further used in NAGPHORMER and LARGEGT. For all experiments, we use the best hyper-parameters outlined in the respective baseline code-bases wherever available for each dataset. Otherwise, standard tuning is done for key hyper-parameters including the number of layers, embedding size, and number of virtual nodes where applicable. We have updated this in Appendix section D.2 of revised manuscript. We select a number of negative features per node using hyperparameter tuning over the range 1-10.

## E  Additional Results

### E.1  Ablation Study

We design 5 variants of NEUTAG, **1)**NEUTAG (COMPLETE) which is the entire architecture **2)** NEUTAG (LOCAL NBRS.), which only consists of attention with local neighbors **3)**NEUTAG$^+$ consists of attentions with local neighbors, feature nodes and feature to feature attention path except attention with negative features **4)**NEUTAG-F2F consists of all attention paths except feature to feature attention and **5)**NEUTAG$^+$-F2F consists of local neighbor attention and feature node attention. Table 6 demonstrates the effectiveness of all the variants. Specifically, we observe that computing node representation by only attending to local neighbors NEUTAG (LOCAL NBRS) results in sub-optimal performance across all datasets. The performance drop is much more significant in non-homophilic graphs Actor and Chameleon. This signifies the importance of various attention paths involving feature nodes in learning both homophilic and heterophilic biases in NEUTAG.

Table 6: Ablation of NEUTAG on node classification task

| Neutag Variants | Cora | CiteSeer | Actor | Chameleon |
|---|---|---|---|---|
| NEUTAG (COMPLETE) | $87.26 \pm 2.14$ | $76.00 \pm 0.99$ | $\mathbf{36.25 \pm 2.43}$ | $\mathbf{65.26 \pm 2.43}$ |
| NEUTAG (LOCAL NBRS.) | $81.01 \pm 3.67$ | $75.49 \pm 1.10$ | $25.52 \pm 0.87$ | $30.70 \pm 1.49$ |
| NEUTAG$^+$ | $87.19 \pm 0.96$ | $\mathbf{77.68 \pm 1.9}$ | $34.93 \pm 0.83$ | $64.07 \pm 2.73$ |
| NEUTAG $-$F2F | $87.12 \pm 1.46$ | $75.70 \pm 0.3$ | $34.26 \pm 1.85$ | $63.02 \pm 3.79$ |
| NEUTAG$^+$$-$F2F | $\mathbf{87.67 \pm 1.10}$ | $74.65 \pm 0.95$ | $34.93 \pm 0.46$ | $64.12 \pm 1.95$ |

Table 6 also indicates that attention with local neighbors and feature nodes (NEUTAG$^+$$-$F2F) is competitive across all datasets. In contrast, additional attention with negative feature nodes and feature-to-feature attention NEUTAG (COMPLETE) provides a performance boost in heterophilic graph Actors and Chameleon.

Moreover, we also perform ablation of NEUTAG with respect to the number of negative feature nodes sampled per graph node to analyze the sensitivity of NEUTAG with respect to the number of negative feature samples. Figure 4 shows that performance improves with increasing number of negatives, but it saturates quickly at around 5.

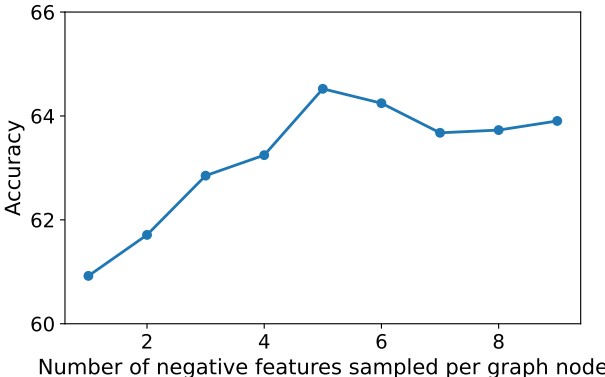

Figure 4: Impact of number of negative feature sampling on model performance on Chameleon dataset

## E.2 Additional Challenging Heterophilic datasets

We further benchmark NEUTAG on challenging heterophilic datasets, with graph statistics and results summarized in Table 7. As shown, NEUTAG consistently outperforms scalable graph transformers.

Table 7: Comparison of NEUTAG with scalable GT on additional challenging heterophilic graphs (Luan et al., 2024a)

| Dataset | # Nodes | # Edges | $\mathbf{H}_{edge}$ | # Features | Avg. feat./ node | NAGPHORMER | GOAT | LARGEGT | NEUTAG |
|---|---|---|---|---|---|---|---|---|---|
| Facebook | 4039 | 88234 | 0.5816 | 1283 | $\approx 8$ | $59.91 \pm 1.11$ | Error | Error | $\mathbf{63.09 \pm 1.29}$ |
| Cornell | 183 | 295 | 0.2983 | 1703 | $\approx 94$ | $58.90 \pm 5.23$ | $67.02 \pm 8.41$ | $53.51 \pm 11.89$ | $\mathbf{77.834 \pm 7.33}$ |
| Squirrel | 5201 | 217073 | 0.2234 | 2089 | $\approx 18$ | $38.05 \pm 2.00$ | $33.56 \pm 0.74$ | $36.5 \pm 2.69$ | $\mathbf{50.36 \pm 2.12}$ |
| Wisconsin | 251 | 499 | 0.1703 | 1703 | $\approx 96$ | $58.42 \pm 4.01$ | $74.11 \pm 7.76$ | $69.01 \pm 7.48$ | $\mathbf{78.034 \pm 4.19}$ |
| Texas | 183 | 309 | 0.0615 | 1703 | $\approx 83$ | $60.61 \pm 7.15$ | $68.64 \pm 4.09$ | $68.10 \pm 10.99$ | $\mathbf{82.69 \pm 4.04}$ |

## E.3 Impact of missing features on Neutag

We conduct two studies to examine how missing features affect NEUTAG. The first study looks at missing features only during the graph transformation stage. Here, the original input features are intact, but node–feature edges are randomly removed to simulate different levels of feature sparsity.

Table 8: Comparison of Neutag with scalable GT on additional NagPhormer datasets

| Dataset | # Nodes | # Edges | $\mathbf{H}_{edge}$ | # Features | Avg. feat./ node | NagPhormer | Goat | LargeGT | Neutag |
|---------|---------|---------|---------------------|------------|------------------|-------------|------|---------|--------|
| Pokec | 1.6M | 30.62M | 0.4449 | 65 | $\approx 20$ | $71.55 \pm 2.40$ | Error | $70.70 \pm 0.21$ | $\mathbf{71.97 \pm 0.22}$ |
| Photo | 7850 | 238163 | 0.8272 | 745 | $\approx 259$ | $94.32 \pm 0.52$ | $\mathbf{95.58 \pm 0.45}$ | $93.76 \pm 0.89$ | $94.98 \pm 0.39$ |
| Computer | 13752 | 491722 | 0.7772 | 767 | $\approx 267$ | $88.90 \pm 0.70$ | $\mathbf{91.55 \pm 0.59}$ | $87.39 \pm 1.10$ | $90.50 \pm 0.32$ |
| CoraFull | 19793 | 126842 | 0.5670 | 8701 | $\approx 57$ | $70.03 \pm 0.91$ | $69.21 \pm 0.64$ | $63.25 \pm 0.65$ | $\mathbf{72.62 \pm 0.44}$ |

We test this on the Chameleon dataset. Dropping feature connections reduces the number of feature–node edges, which lowers the homophily of the transformed graph $\mathcal{G}^{meta}$. This weakens the structural connectivity of the transformed graph and, as expected, leads to a drop in classification accuracy, as shown in the table below.

| Feature Drop Rate in $\mathcal{G}^{meta}$ | Homophily$_{ppr}^{\mathcal{G}}$ | Homophily$_{ppr}^{\mathcal{G}^{meta}}$ | Accuracy |
|-------------------------------------------|---------------------------------|----------------------------------------|----------|
| 0% | 0.2349 | 0.2807 | $65.26 \pm 2.74$ |
| 50% | 0.2349 | 0.2615 | $60.51 \pm 2.15$ |
| 90% | 0.2349 | 0.2398 | $58.45 \pm 2.26$ |

Table 9: Effect of feature drop rate during graph transformation on homophily and accuracy.

In another study, for completeness, we now randomly drop features with varying probability ($p$) from the input node feature matrix itself and benchmark it against the baseline methods in table 10. We observe that Neutag maintains relatively strong performance even under severe feature dropout, suggesting robustness to missing input features. This is an encouraging result. That said, we believe a comprehensive study on robustness under various real-world noise and corruption settings is a significant task that warrants a separate investigation.

### E.4 Training Time and Memory Analysis

To empirically understand the scalability of Neutag, we measure the training time and GPU memory of Neutag vs other scalable GT. The following table 11 reports the training time across large-scale datasets Snap-patents, Pokec, and Arxiv.

Among baselines, Neutag emerges as the second-fastest, and NagPhormer is the fastest to compute attention over layers, while other methods, including Neutag, compute attention over nodes.

Moreover, Table 12 shows the parameter sizes of Neutag and the baselines on the large-scale snapshot-patent dataset. All models are lightweight in terms of parameter size.

### E.5 Comparison of Neutag with Dense-Attention

To empirically evaluate how well Neutag approximates dense attention, we compare it with GraphGPS without the Gnn component (GraphGPS-Gnn). Removing the Gnn module results in a model that relies solely on transformer-based global attention over nodes, effectively working as a dense-attention graph transformer. Table 1 reports the performance comparison. For clarity, we reproduce the relevant results here in table 13. As shown, Neutag achieves comparable or better results, empirically supporting the theoretical analysis presented in theorem 2.

### E.6 Analysis of Structural vs. Feature-based Attention Paths

To better understand how Neutag combines structural information from local neighborhood 9 and global information from feature nodes 11 based on the underlying graph characteristics for node classification, we analyze the relative contribution of each component to the final node representation 15. For this analysis, we consider the linear projection component $\text{MLP}_1^L(\mathbf{H}_{\mathcal{V}:local}^l \mid \mathbf{H}_{\mathcal{V}:+}^l)$ in eq. 15, rewriting it for a graph node $v \in CV$ as follows.

| Method | $p = 0$ | $p = 0.5$ | $p = 0.9$ |
|--------|---------|-----------|-----------|
| GRAPHSAGE | $48.95 \pm 3.16$ | $44.42 \pm 2.04$ | $37.50 \pm 2.81$ |
| GAT | $44.74 \pm 3.29$ | $38.33 \pm 3.73$ | $34.20 \pm 1.46$ |
| GIN | $32.68 \pm 3.68$ | $31.22 \pm 1.02$ | $30.26 \pm 3.54$ |
| MIXHOP | $47.68 \pm 2.89$ | $38.13 \pm 5.12$ | $31.14 \pm 3.54$ |
| LINKX | $48.20 \pm 3.31$ | $42.19 \pm 1.58$ | $35.26 \pm 3.04$ |
| GRAPHGPS | $42.88 \pm 1.88$ | $36.14 \pm 2.73$ | $32.96 \pm 3.77$ |
| EXPHORMER | $45.17 \pm 2.56$ | $42.45 \pm 1.60$ | $35.43 \pm 1.04$ |
| NAGPHORMER | $59.97 \pm 1.72$ | $56.72 \pm 1.90$ | $58.56 \pm 1.31$ |
| GOAT | $53.28 \pm 2.48$ | $43.85 \pm 1.93$ | $34.56 \pm 2.50$ |
| LARGEGT | $57.19 \pm 1.89$ | $55.24 \pm 2.45$ | $52.58 \pm 1.70$ |
| NEUTAG | $\mathbf{65.26 \pm 2.43}$ | $\mathbf{60.40 \pm 0.99}$ | $\mathbf{58.75 \pm 2.06}$ |

Table 10: Performance of different methods under varying feature drop rates ($p$).

| Dataset | N(Millions) | M (Millions) | NAGPHORMER | GOAT | LARGEGT | NEUTAG |
|---------|-------------|--------------|------------|------|---------|--------|
| Snap-patents | 2.92 | 13.97 | 1.1 | $> 24$ | 14 | 6 |
| Pokec | 1.62 | 30.62 | 0.56 | ERROR | 5.5 | 4.2 |
| Arxiv | 0.16 | 1.16 | 0.086 | 7.7 | 0.6 | 1.5 |

Table 11: Training time comparison of NEUTAG with baselines (Hrs.)

$$\mathbf{h}_v^{final} = \mathbf{W} \cdot [\mathbf{h}_v^{struct} \mid \mathbf{h}_v^{global}] \tag{45}$$

where $\mathbf{h}^{struct}$ denotes the representation aggregated from graph neighbors $\mathcal{V} : local$, and $\mathbf{h}^{global}$ denotes the representation aggregated from feature nodes $\mathcal{V} : +$. Now weight matrix $\mathbf{W}$ in eq. 45 can be decomposed as follows.

$$\mathbf{W} = [\mathbf{W}_{struct} \ \mathbf{W}_{global}] \tag{46}$$

Thus, the eq. 45 can be expressed as:

$$\mathbf{h}_v^{final} = \mathbf{W}_{struct}\mathbf{h}_v^{struct} + \mathbf{W}_{global}\mathbf{h}_v^{global} \tag{47}$$

Now, to quantify the contribution of local neighbors(structural) and global information, we measure the magnitude of the corresponding projected representation. For a graph node $v \in \mathcal{V}$, let us define,

$$C_v^{struct} = \|\mathbf{W}_{struct}\mathbf{h}_v^{struct}\|_2, \quad C_v^{global} = \|\mathbf{W}_{global}\mathbf{h}_v^{global}\|_2 \tag{48}$$

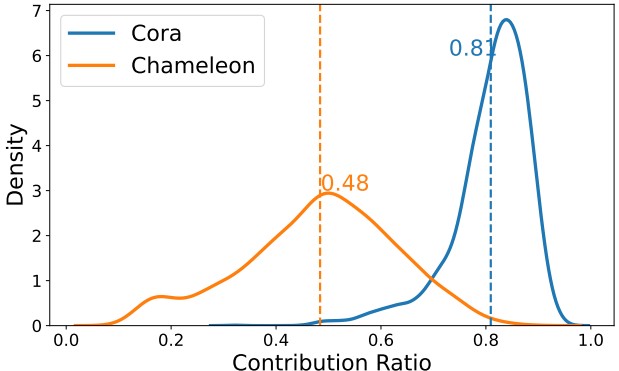

Figure 5: Local neighborhood vs. feature-based attention contribution

| NEUTAG | LARGEGT | GOAT | NAGPHORMER |
|--------|---------|------|------------|
| 742361 ( 3 MB) | 403462( 1.6 MB) | 442125( 1.77 MB) | 176519( 706 kb) |

Table 12: Parameter sizes of NEUTAG and baselines on Snap-patents dataset

| Method | Cora | CiteSeer | Actor | Chameleon | OGBN-Arxiv | OGBN-Arxiv(Year) | Snap-patents |
|--------|------|----------|-------|-----------|------------|------------------|--------------|
| GRAPHGPS-GNN | $72.47 \pm 1.87$ | $71.59 \pm 2.43$ | $37.10 \pm 1.11$ | $47.36 \pm 2.22$ | OOM | OOM | OOM |
| NEUTAG | $\mathbf{87.67 \pm 1.10}$ | $\mathbf{77.68 \pm 1.90}$ | $36.21 \pm 1.2$ | $\mathbf{65.26 \pm 2.43}$ | $70.63 \pm 0.29$ | $\mathbf{53.96 \pm 0.38}$ | $63.00 \pm 0.22$ |

Table 13: Comparison of NEUTAG with respect to dense-attention

Consequently, we define the relative contribution of the structural component as follows for $v \in \mathcal{V}$:

$$R_v^{struct} = \frac{C_v^{struct}}{C_v^{struct} + C_v^{global}} \tag{49}$$

We analyze the distribution of $R_v^{struct}$ across all nodes $\mathcal{V}$ in $\mathcal{G}$. We visualize this distribution for both homophilic graph Cora and heterophilic graph Chameleon in figure 5. Figure 5 clearly shows that NEUTAG primarily relies on local neighborhood with a concentrated distribution with mean of 0.81.

In contrast, on heterophilic Chameleon, the contribution distribution is highly balanced with mean $\approx 0.53$ and shows high variance. This suggests that the model is able to leverage feature-based connections at the node level, rather than follow a fixed structural bias and use feature-based connections as needed. These observations demonstrate that NEUTAG can dynamically adapt to both homophilic and heterophilic graphs by utilizing and balancing both local neighborhood information and feature-based information.

### E.7 Sensitivity to feature binarization for real-valued attributed graphs

We analyze the sensitivity of NEUTAG to the encoding dimension $d'$ used in the $k$-sparse encoder for binarizing real-valued features (Appendix C).

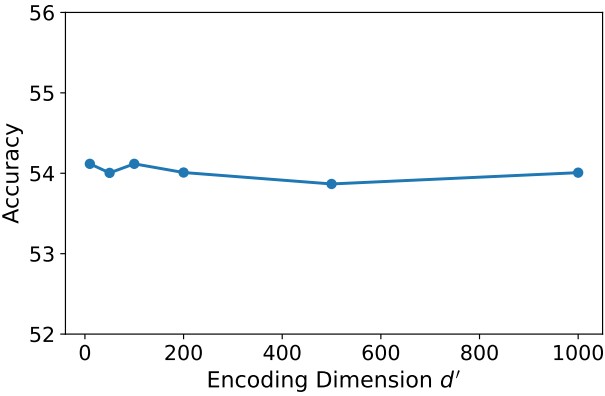

Figure 6: Sensitivity analysis of NEUTAG wrt. hyper-parameter $d'$ used in $k$-sparse encoder for binarization of real-valued attributed features.

Figure 6 shows the performance across a wide range of $d' \in \{10, 50, 100, 200, 500, 1000\}$ with fixed $k = 10$. We observe that the downstream accuracy remains largely stable, with negligible variation across different values of $d'$. This indicates that the proposed binarization scheme is robust to the choice of encoding dimension and does not require careful tuning.

Similarly, we further investigate the sensitivity of NEUTAG to the sparsity hyperparameter $k$ , which determines the top-$k$ largest activations to be annotated as 1 and the rest as 0 during binarization. Figure 7 shows the performance of NEUTAG across a range of $k$. The performance improves as $k$ increases from a very small value, showing that information loss in under-sparse representations, and it saturates with moderate values of $k$, suggesting NEUTAG is not highly sensitive once sufficient $k$ is achieved.

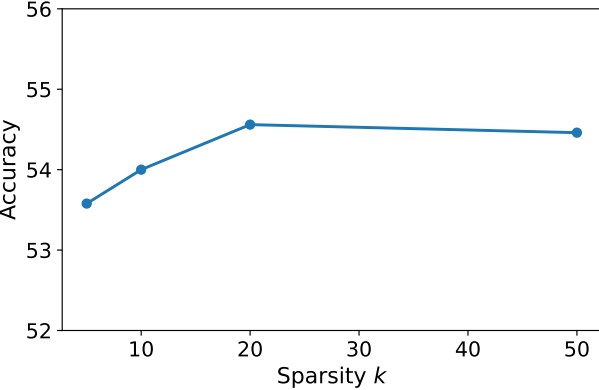

Figure 7: Sensitivity analysis of NEUTAG wrt. sparsity hyper-parameter $k$ in $k$-sparse encoder for binarization of real-valued attributed features.

