# OpenReview forum: "NEUTAG: Graph Transformer for Attributed Graphs"
_TMLR — Accepted by TMLR_

### Review · Reviewer_pVrr · 2026-03-09

**Summary Of Contributions:**

This paper proposes NEUTAG, a sparse graph transformer for node classification task that transforms the input graph into a bipartite structure by decoupling node features into sparse virtual feature nodes. This transformation enables information to flow between distant nodes through shared feature connections, effectively approximating all-node-pair message passing without incurring quadratic computation. The authors provide theoretical analysis on connectivity increase, homophily improvement, and dense attention approximation. Experiments are conducted on seven datasets spanning homophilic, heterophilic, and large-scale graphs.

Strengths:

S1) The core idea of using features as virtual nodes to bridge distant graph nodes is intuitive and elegant. NEUTAG’s architecture is data-agnostic and requires no learned clustering or tunable virtual node counts.

S2) The experimental evaluation covers comprehensive graph types, and NEUTAG demonstrates consistently competitive performance across all settings.

Weaknesses and Corresponding Questions:



W1) The graph transformation assumes binary feature matrices, but many real-world graphs have continuous features. The extension in Appendix C leverages knowledge distillation with a k-sparse encoder to binarize features, introducing an additional training stage and hyper-parameters.

Q1: How sensitive is NEUTAG’s performance to the quality of binarization? A more systematic analysis of how those binarization hyper-parameters affect downstream accuracy is encouraged.

w2) In subsections 3.2 and 3.3, given the F-dimensional feature space, the complexity analysis claims a reduction from quadratic to linear in the number of features F. However, when F is large, the number of feature nodes and associated edges could become substantial.   The empirical wall-clock time and GPU memory occupation statistics on large-scale datasets are encouraged to support the claimed reduction.

w3) Theorem 2 shows that NEUTAG can approximate PERFORMER, which itself approximates full softmax attention. The practical gap between NEUTAG’s attention and dense attention is not empirically quantified.   Empirical results on approximation error measurements are encouraged to demonstrate how closely NEUTAG’s attention could approximate dense attention in practice.

**Audience:**

Yes

**Audience Explanation:**

Graph transformers for node classification tasks are an active and practically important research area, particularly given the scalability challenges of applying transformers to large graphs. This paper’s proposal to use features as virtual nodes provides one perspective that could inspire future work on scalable sparse attention mechanisms. The empirical findings regarding the limitations of hybrid GNN-transformer architectures on heterophilic graphs are also insightful for researchers to deploy graph transformers in diverse settings.

**Claims And Evidence:**

Yes

**Claims Explanation:**

This paper’s claims are demonstrated by comprehensive empirical evidence.

**Requested Changes:**

Please address those questions mentioned in the subsections of weaknesses.

---

> ### Author Response · Authors · 2026-04-12
> **Response to Reviewer pVrr**
>
> We appreciate the reviewer’s constructive comments on our work. Below, we provide our responses to the queries raised. The changes in the revised manuscript have been highlighted in blue. We remain open to addressing any further concerns to clarify the merits of our work.
>
> >**The graph transformation assumes binary feature matrices, but many real-world graphs have continuous features. The extension in Appendix C leverages knowledge distillation with a k-sparse encoder to binarize features, introducing an additional training stage and hyper-parameters.
> >How sensitive is NEUTAG’s performance to the quality of binarization? A more systematic analysis of how those binarization hyper-parameters affect downstream accuracy is encouraged.**
>
> **Response**
>
> We have added a detailed analysis in Appendix E.7 of revised manuscript to study the sensitivity of NEUTAG to binarization process using k-sparse encoder. Specifically, we analyze the impact of key hyper-parameters, including encoding dimension $d'$ and sparsity $k$. This analysis broadly shows that sparsity parameter $k$ has a noticable impact on peformance, while NEUTAG remains relatively stable across encoding dimension $d'$.
>
> >**In subsections 3.2 and 3.3, given the F-dimensional feature space, the complexity analysis claims a reduction from quadratic to linear in the number of features F. However, when F is large, the number of feature nodes and associated edges could become substantial. The empirical wall-clock time and GPU memory occupation statistics on large-scale datasets are encouraged to support the claimed reduction.**
>
> **Response**
>
> We thank the reviewer for highlighting this important point. Dense-attention graph transformers incur $O(N^2)$ complexity, which becomes prohibitive for large-scale graphs as shown in Table 1 of the manuscript, where methods such as GraphGPS and Exphormer fail to scale to large-scale datasets like OGBN-Arxiv and Snap-Patents.
>
> In contrast, NEUTAG projects attention from the node space to the feature space, reducing the complexity to $O(NF)$ thereby avoiding explicit all-pairs node interactions. In large-scale graphs, where $F \ll N$, NEUTAG is able to scale effectively as shown in Table 1 of the manuscript.
>
> We have now added a complexity discussion of NEUTAG in Section 3.3 of the revised draft which shows that the effective computational cost in NEUTAG is governed by the number of node-feature edges, which depends on the number of features applicable per graph node and is typically sparse. So, we have further specified average features per graph node in the dataset statistics Tables 5 and 7 in the Appendix, referenced in Section 4.2 of the main paper.
>
>
> Based on this analysis, while we agree with reviewer when $F \sim N$, the computational advantage diminishes. But this happens because of feature to feature attention paths $O(F^2)$, not because of node-feature interactions which remains sparse as each node is connected to small subset of features. However, this issue is not applicable for large-scale graphs where $N\gg F$.
>
>
> In response to the reviewer’s suggestion, we now report training times (in hours) on the three largest datasets used in the manuscript. Since dense-attention-based graph transformers do not scale to these settings (as shown in Table 1), we compare NEUTAG against scalable graph transformer baselines.
>
> NagPhormer is the fastest since it does not compute attention wrt nodes. It computes attention wrt each layer and #layers$\ll$#nodes. Notably, NagPhormer is also the weakest in terms of accuracy across all baselines.
>
> |Dataset| N (Millions)| M (Millions)| F| NAGPhormer|GOAT|LargeGT| NEUTAG|
> |----|----|----|----|----|----|----| ---|
> |Snap-Patents|2.92 |13.97 | 269|1.1| >24 | 14| 6 |
> |Pokec|1.62 |30.62 |65| 0.56 |ERROR | 5.5 |4.2|
> |OGBN-Arxiv|0.16 |1.16| 128|0.086|7.7|0.6|1.5|
>
> The GPU memory consumption is dictated by the model parameter size. We present the parameter sizes of Neutag and baselines below on large-scale snap-patent dataset.
>
> |  NEUTAG|LargeGT   |  GOAT | NAGPhormer|
> |  ------- | ------- | --- | --- |
> | 742,631  (~3MB)  | 403,462 (~1.6 MB)   | 442,125 (~1.77MB) | 176,519 (~706kb)|
>
>
> While NEUTAG is not the fastest or smallest model among scalable graph transformers, it scales effectively while maintaining consistent performance across diverse settings, including small- and large-scale datasets as well as homophilic and heterophilic graphs, whereas other scalable baselines exhibit more variable performance across datasets(Table 1).
>
> We thank the reviewer for highlighting this, we have included this discussion in section E.4 of Appendix of revised manuscript, referenced in Section 4.4 of main paper.

---

> > ### Author Response · Authors · 2026-04-12
> > **Response to Reviewer pVrr**
> >
> > >**Theorem 2 shows that NEUTAG can approximate PERFORMER, which itself approximates full softmax attention. The practical gap between NEUTAG’s attention and dense attention is not empirically quantified. Empirical results on approximation error measurements are encouraged to demonstrate how closely NEUTAG’s attention could approximate dense attention in practice.**
> >
> > **Response**
> >
> > We thank the reviewer for this suggestion. Directly measuring the approximation error between NEUTAG attention and dense softmax attention is non-trivial especially for large-scale graphs as dense-attention is computationally infeasible.
> > To empirically evaluate how well NEUTAG approximates dense attention, we compare it with GraphGPS on small-scale graphs without the GNN component (GraphGPS-GNN). Removing the GNN module results in a model that relies solely on transformer-based attention over all nodes, effectively working as a dense-attention graph transformer. Following table shows this comparision.
> >
> > | Method              | Cora              | CiteSeer           | Actor             | Chameleon          | OGBN-Arxiv        | OGBN-Arxiv (Year) | Snap-patents     |
> > |--------------------|-------------------|--------------------|-------------------|--------------------|-------------------|-------------------|------------------|
> > |GraphGPS| 83.65 ± 2.67|76.25 ± 1.34|34.30±0.45|42.87±1.88| OOM| OOM | OOM|
> > | GraphGPS-GNN       | 72.47 ± 1.87      | 71.59 ± 2.43       | 37.10 ± 1.11      | 47.36 ± 2.22       | OOM               | OOM               | OOM              |
> > | NEUTAG             | **87.67 ± 1.10**  | **77.68 ± 1.90**   | 36.21 ± 1.20      | **65.26 ± 2.43**   | **70.63 ± 0.29**      | **53.96 ± 0.38**  | **63.00 ± 0.22**     |
> >
> > As shown, NEUTAG achieves comparable or better results across datasets, including large-scale settings where dense-attention becomes infeasible. It provides emperical evidence that NEUTAG can effectively capture similar global dependencies in practice. Moreover, we have added an visualization of importance assigned to local neighborhood vs feature-nodes driven global neighborhood in appendix E.6, referenced in Section 4.4 of main paper which shows that NEUTAG is able to adapt dynamically based on the input graph characterizations such as homophily or heterophiliy.
> >
> > We have included this discussion in Section E.5 of Appendix in revised manuscript and we thank the reviewer for highlighting this.
> >
> >
> > We hope these changes address the reviewer's concern, and we remain open to further feedback on these revisions.

---

### Review · Reviewer_Rdiz · 2026-03-16

**Summary Of Contributions:**

The manuscript introduces NEUTAG for node classification on large-scale attributed graphs. The authors propose transforming the input graph into a bipartite metamorphosis form by introducing virtual nodes corresponding to the deterministic feature set. This structure bypasses standard $\mathcal{O}(N^2)$ dense attention by utilizing an attention mechanism over local neighborhoods and feature nodes, effectively reducing computational complexity.

**Audience:**

Yes

**Audience Explanation:**

- The structural transformation to a bipartite metamorphosis form using deterministic feature-based virtual nodes avoids the parameter overhead and assumptions inherent to clustering-based virtual nodes.
- The method successfully avoids the redundant dependency on GNN modules present in hybrid architectures like GRAPHGPS and EXPHORMER.
- Theorem 1 provides a sound theoretical basis for how feature-node connections exponentially increase L-hop neighborhood reachability.

**Claims And Evidence:**

Yes

**Claims Explanation:**

The stated computational efficiency heavily depends on projecting $\mathcal{O}(N)$ complexity to $\mathcal{O}(F)$.

**Requested Changes:**

- The stated computational efficiency heavily depends on projecting $\mathcal{O}(N)$ complexity to $\mathcal{O}(F)$. The manuscript lacks an explicit discussion on the operational upper bounds of $F$ where the claimed scalability holds.
- The core methodology relies on discrete binary feature presence, as defined by $X[v,f]=1$.However, many standard real-world benchmarks rely on continuous embeddings.
- Theorem 2 posits that NEUTAG can approximate the PERFORMER self-attention layer in at most 4 proposed attention layers. The manuscript needs to clarify whether this 4-layer requirement aligns with the actual hyperparameter depth used in the empirical evaluations. If the experiments use fewer layers, the approximation guarantee may not hold in practice.
- The authors propose degree-based sampling for negative feature edges to maintain efficiency due to the high volume of absent features. However, there is no discussion or ablation regarding the model's sensitivity to the number of negative samples drawn per node.
- The manuscript argues that decoupling features into virtual nodes allows NEUTAG to naturally handle both homophilic and heterophilic graphs without adopting the biases of a local GNN. Therefore, the evaluation should include an analysis or visualization of the learned attention weights.

---

> ### Author Response · Authors · 2026-04-12
> **Response to Reviewer Rdiz**
>
> We appreciate the reviewer’s constructive comments on our work. Below, we provide our responses to the queries raised. The changes in the revised manuscript have been highlighted in blue. We remain open to addressing any further concerns to clarify the merits of our work.
>
> >**The stated computational efficiency heavily depends on projecting complexity O(N) to O(F).The manuscript lacks an explicit discussion on operational upper bounds of F where claimed scalability holds.**
>
>
> **Response**
>
> We thank the reviewer for raising this point. We have added a detailed discussion of computational complexity in Section 3.3 of the revised manuscript.
>
> Specifically, we have added an explicit characterization of the operational complexity and explain the per-layer complexity of each component in  NEUTAG. Further, we clarify that the worst-case complexity is $O(NFd)$, which occurs when all nodes are connected to all feature nodes. We further explain that the scalability benefits are most significant when number of feature nodes is much smaller than number of graph nodes $(F << N)$, which is common in large-scale graphs such as Snap-patents where number of nodes are in millions facilitating NEUTAG to scale effectively.
>
> We now explicitly note that for small-scale graph where $F \approx O(N)$, the computational advantage of NEUTAG over dense-attention diminishes with complexity approaching $O(N^2d)$. However, NEUTAG maintains strong performance as shown in table 1 over small datasets due to its ability to model both local neighbourhood and global neighbourhood with feature-based attention paths.
>
> We hope this clarifies the reviewer's concern.
>
>
> >**The core methodology relies on discrete binary feature presence, as defined by $X[v,f]=1.However, many standard real-world benchmarks rely on continuous embeddings.**
>
> **Response**
>
> While the formulation in Subsection 3.1 assumes binary features, NEUTAG is not restricted to discrete features. In Appendix C of the manuscript, we  proposed a lightweight distillation with k-sparse encoding to convert continuous features into binary attributes, ensuring that the proposed transformation remains applicable. We have now added a more explicit reference to this at the end of Section 3.3. Moreover, datasets in our experiments, such as OGBN-Arxiv and OGBN-Arxiv (year), **already include continuous-valued attributes**.
>
> >**Theorem 2 posits that NEUTAG can approximate the PERFORMER self-attention layer in at most 4 proposed attention layers. The manuscript needs to clarify whether this 4-layer requirement aligns with the actual hyperparameter depth used in the empirical evaluations. If the experiments use fewer layers, the approximation guarantee may not hold in practice.**
>
> **Response**
>
> We thank the reviewer for this important clarification. The "at most $4$ layer result" in theorem 2 is a theoretical expressivity result, showing NEUTAG can approximate a PERFORMER-style attention. The result establishes that such an approximation is *possible* within $4$ layer, but it doesn't imply fewer layers can't learn effective representation. Morever, it also doesn't imply that exactly 4 layers are required in emperical settings. Accordingly, in our experiments, number of layers is treated as a standard hyper-parameter and varies across datasets eg. $2$ layers for Cora and $6$ layers for Chameleon, selected based on validation performance. This is primarly due to downstream node classification task, where type and extent of information required for each node can vary, and may not require aggregating information from entire graph. Since validation performance reflects aggregate behaviour, selected depth typically captures the dominant requirement for the dataset. Moreover, NEUTAG's emperical performance doesn't rely only on explicitly approximating PERFORMER attention. Instead, it comes from its ability to integrate local neighborhood information with global feature-based attention paths. Further, we note that NEUTAG outperforms dense-attention based Graph Transformers such as GraphGPS which we have added as seperate discussion in Appendix E.5.
>
> Finally, Theorem 2 is important as it grounds the proposed architecture within broader landscape of attention mechanisms and motivates its design from theoretical perspective and demonstrate that NEUTAG possesses sufficient expressive power to recover performer-style attention behaviour in principle.
>
> We have also clarified this distinction in section 3.3 of revised draft. We thank the reviewer for highlighting this.

---

> > ### Author Response · Authors · 2026-04-12
> > **Response to Reviewer Rdiz - Part 2**
> >
> > >**The authors propose degree-based sampling for negative feature edges to maintain efficiency due to the high volume of absent features. However, there is no discussion or ablation regarding the model's sensitivity to the number of negative samples drawn per node.**
> >
> > **Response**
> > We have now added a discussion and ablation on number of negative features edges sampled per graph node in Appendix E.1 of the revised manuscript referenced in section 4.4 of the main paper. This analysis shows that performances improves initially as number of negative samples increases but saturates quickly suggesting that relatively small number of negative samples is sufficient.
> >
> > We thank the reviewer for emphasizing this point.
> >
> > >**The manuscript argues that decoupling features into virtual nodes allows NEUTAG to naturally handle both homophilic and heterophilic graphs without adopting the biases of a local GNN. Therefore, the evaluation should include an analysis or visualization of the learned attention weights.**
> >
> > **Response**
> > We have now added a detailed analysis and visualization of the relative contribution local neighbourhood based attention paths and feature-node based attention paths in Section of E.6 of Appendix of revised manuscript referenced in section 4.4 (Comparison with Graph Transformers) of main paper. Specifically, we decompose the final node representation into structural and feature-nodes based components and quantify their relative contributions using norm of their projected representations (Eq. 45-49). We then analyze the distribution of these contributions across nodes.
> >
> > Our analysis shows that on homophilic graphs such as Cora, NEUTAG relies on local neighborhood attention paths with mean relative contribution of 0.81. On the other hand, on heterophilic graphs such as Chameleon, the model exhibits a more balanced reliance between local neighborhood attention paths vs. feature attention paths with mean relative contribution of 0.48.  This result demonstrates that NEUTAG adaptively shifts between attention paths based on underlying graph characteristics.
> >
> > We hope these changes satisfies reviewer's concerns and we remain open to more feedback basis these revisions.

---

### Review · Reviewer_8U44 · 2026-04-06

**Summary Of Contributions:**

This paper proposes NEUTAG, a sparse graph transformer for node classification on attributed graphs. The key idea is to transform the input graph into a “metamorphosis” graph by introducing feature nodes (one per feature dimension/value indicator) and connecting graph nodes to the features they possess. This induces a bipartite augmentation that enables non-local communication via shared features while preserving local graph edges. The model then performs attention over:

$(i)$ local graph neighbors,

$(ii)$ present feature-node neighbors,

$(iii)$ sampled absent feature-node neighbors, and

$(iv)$ feature-to-feature all-pairs attention.

The paper argues that this architecture can approximate all-pairs message passing more efficiently than dense attention, provides theoretical claims about increased connectivity and universal approximation / approximation to Performer-style attention, and evaluates the method on a mix of homophilic and heterophilic node classification benchmarks, including Snap-Patents (2.9M nodes). The reported results show strong and relatively consistent performance across seven datasets, with especially strong gains on Chameleon and large-scale settings.

**Additional Comments:**

Typos and minor suggestions

- On page 5, the term "\mathcal R ^{N \times d}" should be $\mathbb R^{N\times d}

- The way Theorem 2 and 3 are stated is misleading. I think the punctuation and phrasing should be improved.

**Audience:**

Yes

**Audience Explanation:**

Yes. This paper should be of interest to graph ML / graph transformer researchers, especially those working on:

- scalable transformers for node classification,

- heterophilic graphs,

- alternatives to virtual-node / clustering-based sparse attention,

- large attributed graphs.

**Broader Impact Concerns:**

No major ethical concerns beyond standard graph-ML deployment risks. The work is primarily a methodological contribution.

**Claims And Evidence:**

Yes

**Claims Explanation:**

The empirical claim that NEUTAG is a strong and scalable node-classification model on attributed graphs is reasonably supported by experiments across 7 datasets, including large-scale Snap-Patents (2.9M nodes), where it performs strongly and often competitively or best. The ablation also supports that feature-node attention paths are useful, especially on heterophilic graphs.

However, the stronger theoretical and framing claims are not fully convincing:

- the paper claims “assumption-free, data-agnostic” modeling, but the method relies heavily on attributed graphs and explicitly does not apply to non-attributed graphs,

- the “universal approximation” / dense-attention approximation claims appear overstated relative to the evidence presented in the main paper and require much more careful qualification.

- the scalability claim is promising, but the paper lacks runtime and memory comparisons, so the efficiency argument is incomplete despite the large-scale benchmark.

**Requested Changes:**

I thank the authors for their interesting and technically solid work. Below are some questions and comments.

1. Clarification of the theoretical claims:
In particular the “universal approximation” and dense-attention approximation claims; clearly state assumptions and practical scope.

2. Adding runtime and memory benchmarks (peak GPU memory, epoch time, throughput), since scalability is a central claim.

3. Clarifying feature preprocessing, especially for real-valued features (e.g., OGBN-Arxiv), since the main construction is binary/incidence-based and the real-valued extension is only in the appendix.

4. Improving experimental transparency: report actual transformed feature-node sizes / sparsity and provide more details on negative-feature sampling and fairness of baseline tuning.

---

> ### Author Response · Authors · 2026-04-12
> **Response to Reviewer 8U44**
>
> We appreciate the reviewer’s constructive comments on our work. Below, we provide our responses to the queries raised. The changes in the revised manuscript have been highlighted with blue. We remain open to addressing any further concerns to clarify the merits of our work.
>
>
> >**the paper claims “assumption-free, data-agnostic” modeling, but the method relies heavily on attributed graphs and explicitly does not apply to non-attributed graphs**
>
> **Response**
> We thank the reviewer for pointing out this concern with our wording, as the method relies on attributed graphs and is not applicable to non-attributed settings. Our intended meaning was that NEUTAG does not impose additional modeling assumptions, such as the use of homophilic GNN components in graph transformers like GraphGPS, nor does it require designing clustering-based virtual nodes as in GOAT. Instead, NEUTAG derives its attention structure deterministically from the given input features without introducing additional design choices.
>
> To address this concern, we have revised the contribution section of the manuscript by removing the “assumption-free” claim and explicitly clarifying that NEUTAG is designed for attributed graphs. Moreover, we have also added an explicit limitation section 5 in main paper to highlight non-applicability of NEUTAG to non-attributed graphs.
>
> We believe these changes clarify the scope of our method and address reviewer's concern.
>
> >**Clarification of the theoretical claims: In particular the “universal approximation” and dense-attention approximation claims; clearly state assumptions and practical scope.**
>
> **Response**
> We thank the reviewer for raising this clarification.
>
>
> The result in Theorem 2 which states that at most 4 layers are required to approximate PERFORMER style attention is theoretical guarantee. It shows that NEUTAG can approximate a PERFORMER-style attention mechanism and establishes that such approximation is possible with 4 layers, but it doesn't imply that fewer layers can't learn effective representations. In our experiments, the number of layers is treated as a hyperparameter and varies across datasets e.g., 2 for Cora and 6 for Chameleon, selected based on validation performance. This depends on the downstream task, in this case node-classification, where the type and extent of information required can vary across datasets. We have clarified this practical aspect of Theorem 2 in Section 3.3.
>
>
> Similarly, the result in Theorem 3 is intended as theoretical capability result, establishing that NEUTAG's feature-based attention can approximate a dense self-attention layer under the stated assumption. As shown in the proof, this may require $O(N^d)$ parameters, which is impractical for real-world graphs. We have now specified these practical limitation in theorem.
>
>
>
> Primary role of these theorems is to provide theoretical grounding for NEUTAG's feature-based attention mechanism, which does not have a direct analogue in standard transformer settings (e.g., text). Similar to how clustering-based virtual nodes in GOAT can be related to sparse transformer formulations such as Linformer, our theorems connect NEUTAG to Performer-style sparse attention and to dense attention in an approximation-theoretic sense.
>
> In practice, NEUTAG achieves strong performance by leveraging structured attention design that combine local information with feature-based information. Moreover, NEUTAG outperforms dense-attention-based graph transformers on small-scale datasets where such methods are tractable. We have also updated this in revised manuscript in Appendix E.5.
>
> Moreover, we note that Theorem 2 is sufficient to establish the core capabilities of NEUTAG. To avoid overemphasizing theoretical claims that are less relevant in practice, we have moved Theorem 3 from the main text to the appendix and clarified its role as a theoretical result. Moreover, we have made the language of these theorems more formal and precise in revised manuscript.
>
> We thank the reviewer for highlighting this.
> [1]Wang, Sinong, et al. "Linformer: Self-attention with linear complexity." arXiv preprint arXiv:2006.04768 (2020).

---

> > ### Author Response · Authors · 2026-04-12
> > **Response to Reviewer 8U44 Part-2**
> >
> > >**Adding runtime and memory benchmarks (peak GPU memory, epoch time, throughput), since scalability is a central claim.**
> >
> > **Response**
> > We thank the reviewer for highlighting this important point. The scalability of NEUTAG is already demonstrated in the Table 1 of main results, where dense-attention-based graph transformers (e.g., GraphGPS) do not scale to large datasets such as OGBN-Arxiv, OGBN-Arxiv(Year), and Snap-patents. In contrast, NEUTAG scales effectively while maintaining consistent performance across diverse settings, including small- and large-scale datasets as well as homophilic and heterophilic graphs.
> >
> > To further compare NEUTAG with scalable sparse graph transformers, we have now added following results in section E.4 of Appendix of revised manuscript, referenced in Section 4.4 of main paper.
> >
> > We present below the running training times in hours on the three largest datasets used. NagPhormer is the fastest since it does not compute attention wrt nodes. It computes attention wrt each layer and #layers$\ll$#nodes. Notably, NagPhormer is also the weakest in terms of accuracy across all baselines.
> >
> > |Dataset| N (Millions)| M (Millions)| F| NAGPhormer|GOAT|LargeGT| NEUTAG|
> > |----|----|----|----|----|----|----| ---|
> > |Snap-patents|2.92 |13.97 | 269|1.1| >24 | 14| 6 |
> > |Pokec|1.62 |30.62 |65| 0.56 |ERROR | 5.5 |4.2|
> > |OGBN-Arxiv|0.16 |1.16| 128|0.086|7.7|0.6|1.5|
> >
> > The GPU memory consumption is dictated by the model parameter size. We present the parameter sizes of Neutag and baselines below on large-scale snap-patent dataset.
> >
> > |  NEUTAG|LargeGT   |  GOAT | NAGPhormer|
> > |  ------- | ------- | --- | --- |
> > | 742,631  (~3MB)  | 403,462 (~1.6 MB)   | 442,125 (~1.77MB) | 176,519 (~706kb)|
> >
> >
> > >**Clarifying feature preprocessing, especially for real-valued features (e.g., OGBN-Arxiv), since the main construction is binary/incidence-based and the real-valued extension is only in the appendix.**
> >
> > **Response**
> > In Appendix C of the manuscript, we had proposed a lightweight distillation with k-sparse encoding to convert continuous features into binary attributes, ensuring that the proposed transformation remains applicable. We have added a discussion of this in Appendix C.
> >
> > We have now added a more explicit reference to this at the end of Section 3.3 as it was earlier insufficiently highlighted in the main methodology section of manuscript. Moreover, datasets in our experiments, such as OGBN-Arxiv and OGBN-Arxiv (year), **already include continuous-valued attributes.**
> >
> > >**Improving experimental transparency: report actual transformed feature-node sizes / sparsity and provide more details on negative-feature sampling and fairness of baseline tuning.**
> >
> > **Response**
> > We have now added a discussion and ablation on number of negative features edges sampled per graph node in Appendix E.1 of the revised manuscript. We thank the reviewer for emphasizing this point.
> >
> > For all experiments, we use the best hyperparameters outlined in the respective baseline codebases wherever available for each dataset. Otherwise, standard tuning is done for key hyperparameters including the number of layers, embedding size, and number of virtual nodes where applicable. We have updated this in Appendix section D.2 of revised manuscript .
> >
> > We have now added the feature sparsity in the dataset statistics tables 5,6 and 8 in appendix of revised manuscript.
> >
> > <!-- |Dataset| N (Millions)| M (Millions)| F| Average features per node (Feature Sparsity)|
> > |----|----|----|----|----|
> > |Cora|2708|10556|1433|~18|
> > |CiteSeer|3327|9104|3703|~32|
> > |Actor|34493|495924|8415|~5|
> > |Chameleon|7600|33544|931|~13|
> > |OGBN-Arxiv|0.16 |1.16| 128|128|
> > |OGBN-Arxiv (year)|0.16 |1.16| 128|128|
> > |Snap-patents|2.92 |13.97 | 269|5|  -->
> >
> >
> > >**Minor Suggestion**
> > >On page 5, the term "\mathcal R ^{N \times d}" should be $\mathbb R^{N\times d}
> >
> > **Response** We thank the reviewer for pointing this. These have been corrected in revised manuscript.
> >
> > >The way Theorem 2 and 3 are stated is misleading. I think the punctuation and phrasing should be improved.
> >
> > **Response** We have made the language of these theorems more formal and precise in revised manuscript.
> >
> >
> > We hope these changes satisfies reviewer's concerns and we remain open to more feedback basis these revisions.

---

### Decision · Action_Editor_yqsB · 2026-05-31

**Recommendation:** Accept with minor revision

**Additional Comments:**

This submission was reviewed by three expert reviewers. All three reviewers agreed that the paper's main claims are supported by accurate and convincing evidence.

However, the reviewers also raised several concerns and requested a number of revisions. These concerns primarily revolved around the following: (i) the assumption of binary feature matrices and the lack of a thorough evaluation of the proposed binarization procedure for continuous features (e.g., sensitivity analysis); (ii) the absence of runtime and memory benchmarks; (iii) theoretical claims that appeared somewhat overstated; (iv) the connection between the theoretical results and the experimental findings; and (v) the lack of an ablation study evaluating the model's sensitivity to the number of negative samples drawn per node.

The authors addressed most of these concerns in their revision. Following the rebuttal and revision process, all three reviewers recommended "Leaning Accept". Based on the reviews, the authors' responses, and the revised manuscript, I recommend accept with a minor revision. Specifically, I request that the claim that the proposed model is always more efficient than Graph Transformers is revised in the final version of the manuscript to more accurately reflect what is discussed above. I also encourage the authors to address any remaining concerns raised by the reviewers in the final version.

**Audience:**

Yes

**Audience Explanation:**

TMLR's audience includes several researchers working in the field of graph machine learning. Those individuals are likely to be interested in the paper's findings.

**Claims And Evidence:**

No

**Claims Explanation:**

The main claims made in the paper are the following:

- The proposed method is designed as a more time- and memory-efficient alternative to Graph Transformers, which exhibit poor scalability on very large graphs. While this claim is valid for real-world graphs, if the number of features equals the number of nodes and every node is connected to every feature node, the complexity of the proposed method becomes worse than that of standard Graph Transformers. Therefore, this claim is not entirely accurate, and I request the authors to revise it in the final version of the manuscript.

- Under certain conditions, the proposed method is a universal approximator for permutation equivariant sequence-to-sequence functions. The paper provides a proof of this claim that builds upon previously established results.

- The proposed method is competitive across homophilic and heterophilic tasks, as well as on both small-scale and large-scale graphs. This claim is supported by extensive experimental results reported in the manuscript.

---

> ### Author Response · Authors · 2026-06-04
> **Response to Action Editor's Comments**
>
> We thank the Action Editor for the recommendation and the reviewers for their valuable feedback, which has helped improve the quality of our manuscript.
>
> In the camera-ready version, as recommended, we have clarified claims on NEUTAG's scalability. Specifically, we have added a caveat that the scalability advantages of the proposed framework are applicable in the practical setting where the number of connected features per node is substantially smaller than the number of nodes. This clarification has been added in the Abstract, Contributions section, and Methodology section.
>
> Furthermore, the Methodology section contains a dedicated discussion of worst-case complexity, explicitly noting that the computational advantage diminishes when all graph nodes are connected to all feature nodes and $F = \mathcal{O}(N)$.
>
> The remaining concerns were addressed during the rebuttal process and have been incorporated into the revised manuscript and appendix.
>
> We once again thank the Action Editor and the reviewers for their time, effort, and constructive feedback.